# Targeted Attack Improves Protection against Unauthorized Diffusion Customization

**Boyang Zheng**[1]* **Chumeng Liang**[2]* **Xiaoyu Wu**[1]
[1] Shanghai Jiao Tong University, [2] University of Southern California

## Abstract

Diffusion models build a new milestone for image generation yet raising public concerns, for they can be fine-tuned on unauthorized images for customization. Protection based on adversarial attacks rises to encounter this unauthorized diffusion customization, by adding protective watermarks to images and poisoning diffusion models. However, current protection, leveraging untargeted attacks, does not appear to be effective enough. In this paper, we propose a simple yet effective improvement for the protection against unauthorized diffusion customization by introducing targeted attacks. We show that by carefully selecting the target, targeted attacks significantly outperform untargeted attacks in poisoning diffusion models and degrading the customization image quality. Extensive experiments validate the superiority of our method on two mainstream customization methods of diffusion models, compared to existing protections. To explain the surprising success of targeted attacks, we delve into the mechanism of attack-based protections and propose a hypothesis based on our observation, which enhances the comprehension of attack-based protections. To the best of our knowledge, we are the first to both reveal the vulnerability of diffusion models to targeted attacks and leverage targeted attacks to enhance protection against unauthorized diffusion customization.

[Warning: This paper contains images that some readers may find disturbing.]

## 1 Introduction

Diffusion Models (Sohl-Dickstein et al., 2015; Song & Ermon, 2019; Ho et al., 2020; Song et al., 2020; Rombach et al., 2022) has achieved state-of-the-art performance in image synthesis. One distinct advantage of diffusion models is the capability of controlling and customizing the image generation via reference images (Meng et al., 2021), text description (Saharia et al., 2022), sketches (Zhang & Agrawala, 2023), styles (Ruiz et al., 2023; Zhang et al., 2023), and identities (Ye et al., 2023; Wang et al., 2024). However, this power is a double-edged sword, for it also makes possible the unauthorized diffusion customization, where malicious individuals seek interests from customizing diffusion models based on unauthorized images, e.g. copyright photos. This has long been a public concern that draws attention from the whole society (Fan et al., 2023; Chen et al., 2023; Wang et al., 2023b). Countermeasures are needed to protect private images from being arbitrarily used for unauthorized diffusion customization.

Recognizing the need, researchers developed protections based on adversarial attacks on diffusion models (Salman et al., 2023; Liang et al., 2023; Shan et al., 2023a; Van Le et al., 2023; Shan et al., 2023b). These protections add tiny adversarial perturbations to images. Diffusion models customized on these perturbed images will be poisoned, and the generated images will suffer a degradation in quality. This makes unauthorized customization fail in producing usable images, thus penalizing such attempts. Applications based on these protections (Shan et al., 2023a; Liang & Wu, 2023; Shan et al., 2023b) have safeguarded private images in practice.

However, current protections are not strong enough to hold effective when the protection strength is limited, i.e. the adversarial perturbation budget is small, which is a practical need to maintain the visual effects of the protected images. Figure 1 demonstrates examples of customization images with existing protection. Under a small protection strength, existing protections appear to put patterns on

---

*equal contribution

Data  Clean  PhotoGuard PhotoGuard+  AdvDM  ASPL  ACE  ACE+

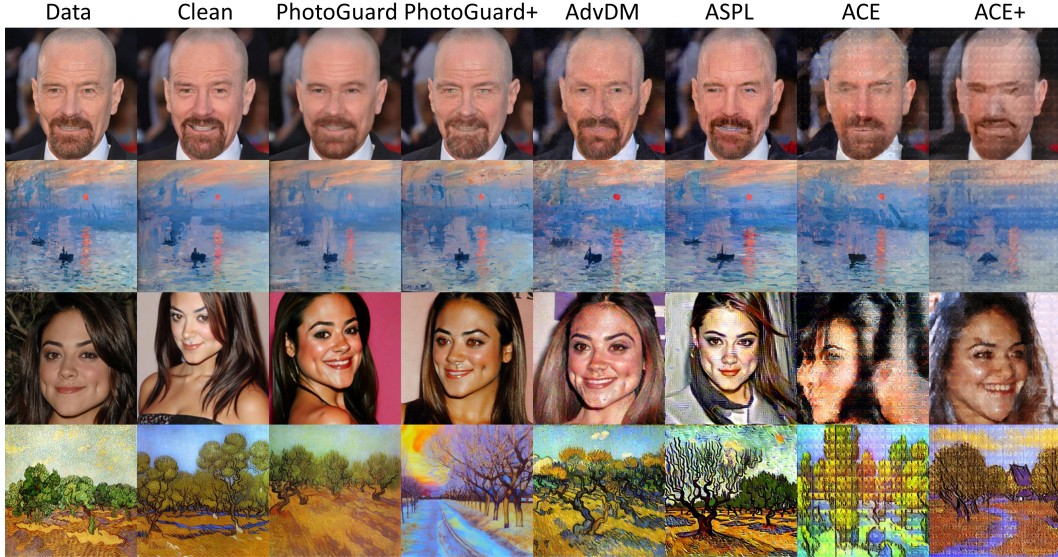

Figure 1: Output images of two mainstream diffusion customization, SDEdit (top two rows) and LoRA (bottom two rows) under different protections with perturbation budget $4/255$. **ACE** and **ACE+** are our targeted attack, while others are baselines based on untargeted attacks.

only some locations of customization output images. The main content of the image remains intact. This raises concerns about the effectiveness of current protections and asks for stronger ones.

In this paper, we propose a stronger protection against unauthorized diffusion customization based on adversarial attacks. We notice that all state-of-the-art protections depend on untargeted attacks on diffusion models. We explore adapting the targeted attack for adversarial attacks on classifiers to the context of diffusion models. Specifically, the targeted attack minimizes the distance between model outputs and a pre-defined target. Since this attack creates score-function errors in a consistent direction, we denote this method by **A**ttacking with **C**onsistent score-function **E**rrors (**ACE**). Empirically, we find that the selection of the target has great impacts on the performance of ACE. With properly selected targets, ACE can produce customization output images all covered by chaotic patterns (see the last two columns of Figure 1). Through extensive experiments, we validate that ACE outperforms all baselines in two mainstream diffusion customizations.

In addition, we delve into the mechanism of targeted and untargeted attacks on diffusion models. Based on our observation, we propose a hypothesis on how attack-based protections work to degrade diffusion customization, that customization diffusion models are degraded because they try to minimize the score-function error introduced by attacks. According to this hypothesis, we show that targeted attacks like ACE unifie the pattern of score-function errors in different protected images, thus reinforcing the bad pattern learned by diffusion models. This provides an intuitive explanation on why the targeted attack ACE outperforms all untargeted baselines.

The contributions of this paper are two-fold. First, we propose ACE, a novel protection method against unauthorized diffusion customization. ACE outperforms existing protections and sets a new milestone in poisoning diffusion models and degrading the image quality of diffusion customization. To the best of our knowledge, ACE is the first targeted-attack-based protection that outperforms untargeted-attack-based protections. Second, we propose a hypothesis on how attack-based protections work based on our observation. This will help future research comprehend and develop even stronger protections against unauthorized diffusion customization.

## 2 BACKGROUND

**Diffusion Models** Diffusion Models (Sohl-Dickstein et al., 2015; Song & Ermon, 2019; Ho et al., 2020) are built on a forward and reverse process that map the noise and data distribution. The forward process perturbs the data progressively to approximate Gaussian noise. The reverse process exploits a parameterized score-function $s_\theta$ to generate data by denoising noise step-by-step. These

two processes can be interpreted by a forward and a reverse SDE (Anderson, 1982; Song et al., 2020):

$$
\begin{aligned}
d\boldsymbol{x} &= \boldsymbol{f}(\boldsymbol{x},t)dt + g(t)d\boldsymbol{w} \\
d\boldsymbol{x} &= [\boldsymbol{f}(\boldsymbol{x},t) - g^2(t)\nabla_{\boldsymbol{x}} \log p_t(\boldsymbol{x})]dt + g(t)d\bar{\boldsymbol{w}}
\end{aligned}
\tag{1}
$$

where $\boldsymbol{w}$ and $\bar{\boldsymbol{w}}$ are standard Wiener processes when time flows from 0 to $T$ and backwards, and $dt$ are infinitesimal timesteps (Song et al., 2020). Diffusion models are then trained by minimizing the *score-function error*, the difference between predicted score-function $s_\theta(x,t)$ by neural network $\theta$ and the ground truth score function $\nabla_{\boldsymbol{x}} \log p_t(\boldsymbol{x})$, since the score function is the only unknown term in Equation 1. The training objective is given by:

$$
\min_\theta \mathbb{E}_t \mathbb{E}_{x(t)|x(0)} \| \underbrace{s_\theta(x(t),t) - \nabla_{x(t)} \log p_t(x(t)|x(0))}_{\epsilon_\theta(x(t),t): \text{ score-function error}} \|_2^2
\tag{2}
$$

Following all existing research on protections against unauthorized diffusion customization (Liang et al., 2023; Liang & Wu, 2023; Shan et al., 2023a;b; Salman et al., 2023; Van Le et al., 2023; Xue et al., 2023), we mainly consider a specific kind of diffusion models, Latent Diffusion Model (LDM) (Rombach et al., 2022), in the context of unauthorized diffusion customization. LDM deploys its forward and reverse processes on a latent representation space $z = \mathcal{E}(x)$ given by an encoder $\mathcal{E}(\cdot)$, and instantiates $\boldsymbol{f}(\boldsymbol{x},t)$ and $g(t)$ as the linear schedule (Ho et al., 2020).

**Diffusion Customization** Diffusion customization (or diffusion personalization (Zhang et al., 2024)) refers to customizing the generated images by diffusion models according to the references from user inputs. These user inputs cover reference images (Meng et al., 2021), text description (Saharia et al., 2022), sketches (Zhang & Agrawala, 2023), styles (Ruiz et al., 2023; Zhang et al., 2023), and identities (Ruiz et al., 2023; Ye et al., 2023; Wang et al., 2024). Among these customization methods, LoRA+DreamBooth (Hu et al., 2021; Ruiz et al., 2023) and SDEdit (Meng et al., 2021) are two mainstream methods that cover most of the customization practice (See our brief survey in Appendix E). Hence, we follow the setup of existing works (Van Le et al., 2023; Liang et al., 2023; Xue et al., 2023) to focus on these two methods.

**Adversarial Attacks & Protections against Unauthorized Diffusion Customization** Adversarial attacks on Diffusion Models (Salman et al., 2023; Liang et al., 2023; Shan et al., 2023a; Liang & Wu, 2023; Van Le et al., 2023; Zhao et al., 2023; Xue et al., 2023; Liu et al., 2024b;a) have been used to protect private images against unauthorized diffusion customization. Specifically, these attacks add tiny adversarial perturbations to private images as protections. Diffusion models customized (fine-tuned (Hu et al., 2021; Ruiz et al., 2023) or referring to (Meng et al., 2021)) the protected images will produce images of degraded quality in customized generation. This will penalize the attempt to customize diffusion models on unauthorized private images.

It is noticeable that mainstream protections are based on untargeted attacks on diffusion models. Those untargeted objectives maximize the score-function error in Equation 2 by optimizing the image perturbation $\eta$ within the perturbation budget $\zeta$, which is formulated by Equation 3.

$$
\max_\eta \mathbb{E}_t \mathbb{E}_{z'(t)|z'(0)} \| \underbrace{s_\theta(z'(t),t) - \nabla_{z'(t)} \log p_t(z'(t)|z'(0))}_{\epsilon_\theta(z'(t),t): \text{ score-function error}} \|_2^2
\tag{3}
$$
$$
\text{where } z'(0) = \mathcal{E}(x'), x' = x + \eta, \eta \in [-\zeta, \zeta]
$$

Here, $x$ is the clean image and $z'(0)$ is the latent representation of $x'$ with LDM encoder $\mathcal{E}(\cdot)$. Empirically, this untargeted attack does not yield satisfying effectiveness in degrading the quality of the customization image (see the examples in Figure 1). In this paper, we improve the protection effectiveness by introducing targeted attacks in Section 3 and provide qualitative analysis to explain our improvements in Section 5.

## 3 Targeted Attack as Protection against Unauthorized Diffusion Customization

Our intuition of introducing targeted attack is simple. We notice that there is an objective mismatch between protection against unauthorized diffusion customization and untargeted attacks on diffusion models (Equation 3). Specifically, protection seeks for **degrading the image quality of diffusion**

**customization**, while untargeted attack's goal is to minimize the log-likelihood of the protected image and make it **not like a real image**. However, images with bad quality are not necessary to have a low log-likelihood. Hence, a more straightforward approach to degrade the quality of the customization image is to directly make the protected image **similar to a bad quality image**. When referring to the protected image, diffusion customization will generate images similar to *that* bad-quality image. We implement this approach by introducing a targeted attack on protection (Section 3.1) and use a bad quality image as the target (Section 3.2).

## 3.1 ATTACKING WITH CONSISTENT ERRORS

We start from comparing classical untargeted and targeted adversarial attacks (Goodfellow et al., 2014; Madry et al., 2017), where both work soundly in attacking classifiers:

$$\begin{cases} \max_{\eta} \|p_\theta(y|x') - \boldsymbol{e}_{true}\|_2^2, & \text{(untargeted attack)} \\ \min_{\eta} \|p_\theta(y|x') - \boldsymbol{e}_{target}\|_2^2, & \text{(targeted attack)} \end{cases}$$

Here, $\boldsymbol{e}_{true}$ and $\boldsymbol{e}_{target}$ are one-hot expressions of the ground-truth label $y_{true}$ and target label $y_{target}$. We notice that targeted attack simply 1) replace the ground truth in untargeted attack with a target and 2) flip maximization to minimization to reduce the distance between the adversarial label and the target label. Inspired by this comparison, we conduct a mirroring transformation on the untargeted attack, given by Equation 3, and transfer it to a targeted attack.

First, we replace the ground truth with a target. The ground truth in Equation 3 refers to the score function $\nabla_{z'(t)} \log p_t(z'(t)|z'(0))$. We replace it with a pre-defined *target* $\mathcal{T}$. Second, we replace maximization with minimization to minimize the distance between the predicted score function $s_\theta(z'(t), t)$ and the target $\mathcal{T}$. This yields a new targeted attack on diffusion models, shown as follows:

$$\begin{cases} \max_{\eta} \mathbb{E}_t \mathbb{E}_{z'(t)|z'(0)} \|s_\theta(z'(t), t) - \nabla_{z'(t)} \log p_t(z'(t)|z'(0))\|_2^2, & \text{(untargeted attack)} \\ \min_{\eta} \mathbb{E}_t \mathbb{E}_{z'(t)|z'(0)} \|s_\theta(z'(t), t) - \mathcal{T}\|_2^2, & \text{(targeted attack)} \end{cases}$$

For different protected images $x'$, this targeted attack uses a consistent target $\mathcal{T}$ as the direction to optimize adversarial perturbations. Hence, we denote this novel targeted attack by **A**ttacking with **C**onsistent **E**rrors (ACE).

$$\min_{\eta} \mathbb{E}_t \mathbb{E}_{z'(t)|z'(0)} \|s_\theta(z'(t), t) - \boldsymbol{\mathcal{T}}\|_2^2$$
$$\text{where } z'(0) = \mathcal{E}(x'), x' = x + \eta, \eta \in [-\zeta, \zeta] \tag{4}$$

Intuitively, ACE misleads the model to predict score function with the pattern of target $\mathcal{T}$ for the protected image $x'$. When customized on $x'$, diffusion models will learn the pattern of $\mathcal{T}$. Therefore, by placing a chaotic pattern in $\mathcal{T}$, we can then make diffusion models that are customized on the protected image learn the chaotic pattern and generate images with this pattern, thus degrading the performance of unauthorized diffusion customization.

In addition to ACE, we also consider an existing targeted attack objective introduced in Liang & Wu (2023), which is specific designed for LDM (Rombach et al., 2022), the point-of-interests diffusion model of unauthorized customization. This term directly minimizes the distance between a target $\mathcal{T}$ and the latent representation of the image $x'$. Empirically, this objective is useful for encountering image reference diffusion customization such as SDEdit (Meng et al., 2021). We therefore weight this objective by $\alpha$ and combine it with ACE, which creates a new attack denoted by ACE+:

$$\min_{\eta} \mathbb{E}_t \mathbb{E}_{z'(t)|z'(0)} \|s_\theta(z'(t), t) - \boldsymbol{\mathcal{T}}\|_2^2 + \alpha \|z'(0) - \boldsymbol{\mathcal{T}}\|_2^2$$
$$\text{where } z'(0) = \mathcal{E}(x'), x' = x + \eta, \eta \in [-\zeta, \zeta] \tag{5}$$

## 3.2 TARGET SELECTION

As mentioned in Section 3.1, good protection effectiveness of ACE/ACE+ depends on a carefully-designed target $\mathcal{T}$. This target must be with chaotic patterns to human vision so that unauthorized diffusion customization will learn these chaotic patterns from the protected image and suffer performance degradation. Our next task is to design such targets with chaotic patterns.

Previous research suggests several aspects of a chaotic patterns from the perspective of human vision: 1) high contrast stripes (Wilkins, 1995), 2) Moire patterns (Amidror, 2009), and 3) overly busy patterns (Bell, 2001). Following these suggestions, we pick the latent representation of the target image used in Liang & Wu (2023) as our basic target. This target has a high contrast of black and white and repeats a motif to create overly busy and Moire-like patterns, which fits the standard of chaotic patterns for human vision. Based on the basic target, we tune the contrast and the repeating number of the motif as hyperparameters. See Appendix B.5.1 for details. With the optimal contrast and repeating number, we determine the final target $\mathcal{T}$ as shown in Figure 2.



Figure 2: Target $\mathcal{T}$ (left) and its corresponding image (right)

To cross-validate our insights in target selection, we additionally design a similar target following the above ideas of chaotic patterns. Specifically, we change the motif of the target in Figure 2 to another, improve its contrast to black and white, and repeat it in one image. This new target is shown in Figure 10. We denote ACE with this new target as ACE* and evaluate ACE* together with ACE and ACE+ in Section 4 as an ablation study.

### 3.3 Implementation as Protection against Unauthorized Customization

Our method combines the objective function in Equation 4/ 5 and the selected target $\mathcal{T}$. We also include fine-tuning steps in our method, following ASPL (Van Le et al., 2023) (5 steps for each iteration). As discussed in its original paper, this is the standard poisoning pipeline for adversarial attacks and brings simple performance gain for adversarial attacks on diffusion models, both untargeted and targeted. For the optimization of ACE/ACE+, we use the classic PGD algorithm (Madry et al., 2017), following existing protections. Putting everything together, we finalize our method in Algorithm 1. To use ACE/ACE+ as protection against unauthorized diffusion customization, one can input the clean image $x$ into Algorithm 1 and take the output image $x'$ as the protected version of $x$. Other implementation details are given in Appendix A.

Note that ACE/ACE+ are the first targeted attack that outperform untargeted attack on diffusion models (see our experiments in Section 4). However, they are not the first targeted attack on diffusion models. While targeted attacks were considered by previous research (e.g. ASPL-T (Van Le et al., 2023)), they failed in beating untargeted attacks. We discuss the reason in Appendix B.5.1.

---

**Algorithm 1** Attacking with Consistent Errors (ACE)

---

1: **Input:** Image $x$, diffusion model $\theta$, learning rates $\alpha, \gamma$, epoch numbers $N, M, K$, budget $\zeta$, diffusion training objective in Equation 2, ACE/ACE+ objective function $J$ in Equation 4 & Equation 5.
2: **Output:** Adversarial example $x'$
3: Initialize $x' \leftarrow x$.
4: **for** $n$ from 1 to $N$ **do**
5:    **for** $m$ from 1 to $M$ **do**
6:       $\theta \leftarrow \theta - \gamma \nabla_\theta \mathcal{L}_{LDM}(x', \theta)$
7:    **end for**
8:    **for** $k$ from 1 to $K$ **do**
9:       $x' \leftarrow x' - \alpha \nabla_{x'} J$
10:       $x' \leftarrow \text{clip}(x', x - \zeta, x + \zeta)$
11:       $x' \leftarrow \text{clip}(x', 0, 255)$
12:    **end for**
13: **end for**
14: **return** $x'$

---

## 4 Experiments

We compare ACE/ACE+/ACE* to existing protections against unauthorized diffusion customization in two main diffusion customization methods, LoRA+DreamBooth (Hu et al., 2021; Ruiz et al., 2023) and SDEdit (Meng et al., 2021). We also evaluate our protection on DreamBooth and Stable Diffusion 3 (Esser et al., 2024) (Appendix B.3). Our methods outperform baselines both in quantitative metrics and qualitative visualization (Section 4.2). We conduct user studies in an artist community, which validates the superiority of our methods (Section 4.3). Furthermore, we test the transferability (Section 4.4) and robustness to purification (Section 4.5) of ACE/ACE+. We investigate the computational cost of our methods and baselines (Appendix B.1). Our ablation studies focus on the effect of the objective function, target selection (Appendix B.5.1), different text prompts (Appendix B.5.2) and perturbation magnitudes (Appendix B.5.3) to the protection effectiveness.

### 4.1 EXPERIMENTAL SETUPS

**Experimental Setups on LoRA+DreamBooth** LoRA+Dreambooth (Hu et al., 2021; Ruiz et al., 2023) (LoRA for simplicity) is the most widely used customization method for diffusion models, occupying a share of more than $90\%$ in the open-source customization platform. LoRA fine-tunes diffusion models with low-ranked adapters on dozens of reference images and generates images with similar contents or styles. Due to its power and convenience, LoRA is the main tool for customization of unauthorized diffusion.

In our experiments, we use LoRA to fine-tune the diffusion model on protected reference images. Fine-tuning is done with 20 protected images with the same content or style. We then generate 100 output images with the fine-tuned customization diffusion models and assess image quality with CLIP-IQA (Wang et al., 2023a) (CLIP-IQA). A high CLIP-IQA score indicates low image quality and strong protection performance. We also follow (Van Le et al., 2023) to introduce two facial image quality metrics in our experiments on CelebA-HQ: Face Detection Failure Rate (FDFR) (Deng et al., 2020) and Identity Score Matching (ISM) (Deng et al., 2019). A strong protection expects a high FDFR by making the face detection (is there a face) failed and a low ISM by disrupting the facial identification (who is the face).

**Experimental Setups on SDEdit** SDEdit (Meng et al., 2021) is an image-to-image customization method that modifies the content or style of a single reference image while keeping the structural similarity, i.e. the layout of the reference image.

In our experiments, we use SDEdit to generate output images based on protected reference images. The strength of SDEdit is set as 0.3. Successful protections destroy this structural similarity between the output image and its corresponding reference image. Hence, we adopt Multi-Scale SSIM (MS-SSIM) (Wang et al., 2003) and CLIP Image-to-Image Similarity (CLIP-SIM) (Wang et al., 2023a), two measurements for image-to-image similarity. We include MS-SSIM for structural similarity and CLIP-SIM for semantic similarity. A strong protection is expected to have both metrics low.

**Datasets & Backbone Model** The experiment is conducted on CelebA-HQ (Karras et al., 2017) and Wikiart (Saleh & Elgammal, 2015). For CelebA-HQ, we select 100 images and each of 20 photos describe one identical person. For Wikiart, we select 100 paintings that each of 20 paintings come from the same artist. We use Stable Diffusion 1.5 as the backbone model for protections, since it is the most popular diffusion model in (unauthorized) diffusion customization. We also investigate the cross-model transferability of ACE/ACE+ with Stable Diffusion 1.4 and Stable Diffusion 2.1 in Section 4.3. As discussed in Section 2, we do not consider other diffusion models because they are either 1) close-sourced so that they do not support customization (e.g. DALLE3) or 2) lack of well implemented customization pipeline (e.g. DeepFloyd IF).

**Hyperparameters** The adversarial budget $\zeta$ is set as $4/255$. Note that this budget is smaller than those in existing literature, such as $8/255$ Liang et al. (2023); Van Le et al. (2023) and $16/255$ Salman et al. (2023), for these large budgets will add perceptible noise to the image that hurt the image quality. Hence, we use a smaller budget to simulate the real-world application scenario. The step length and the number of step in PGD (Madry et al., 2017) are $5 \times 10^{-3}$ and 50. We adopt LoRA for fine-tuning steps in Algorithm 1. Other hyper-parameters are omitted to Appendix A.

**Baselines** We compare ACE/ACE+ with existing methods, including AdvDM (Liang et al., 2023), PhotoGuard (Salman et al., 2023), and ASPL (Anti-DreamBooth) (Van Le et al., 2023). Encoder Attack and Diffusion Attack in PhotoGuard (Salman et al., 2023) are denoted by PG and PG+, respectively. We detail our baseline selection in Appendix A

### 4.2 OVERALL RESULT

We compare ACE/ACE+/ACE$^*$ to existing protections against unauthorized diffusion customization. The overall results of the comparison are given in Table 1, where ACE/ACE+/ACE$^*$ outperform all the baseline methods in degrading the image quality of SDEdit and LoRA. It is noticeable that ACE achieves a FDFR of **1.00**[1]. This means that $0\%$ of the customization images are recognized as face images with the protection of ACE. This is strong evidence that our protections significantly outperform existing methods. Moreover, ACE$^*$ performs similarly or even better than ACE/ACE+,

---

[1] When FDFR is 1.00, ISM is not defined because no face is detected in all the output images

Table 1: Comparison of baseline protections and our protections on LoRA and SDEdit. ACE is our basic method in Equation 4. ACE+ combines ACE and an existing targeted attack (Equation 5). ACE* uses a target other than ACE/ACE+.

| | CelebA-HQ | | | | | WikiArt | | |
| | SDEdit | | | LoRA | | SDEdit | | LoRA |
| | MS-SSIM ↓ | CLIP-SIM ↓ | CLIP-IQA ↑ | FDFR ↑ | ISM ↓ | MS-SSIM ↓ | CLIP-SIM ↓ | CLIP-IQA ↑ |
|---|---|---|---|---|---|---|---|---|
| Clean | 0.88 | 93.38 | 20.66 | 0.02 | 0.69 | 0.62 | 89.77 | 22.88 |
| PG | 0.86 | 89.24 | 27.52 | 0.02 | 0.72 | 0.62 | 88.01 | 37.52 |
| PG+ | 0.82 | 91.00 | 22.91 | 0.04 | 0.71 | 0.57 | 89.80 | 32.62 |
| AdvDM | 0.81 | 83.41 | 24.53 | 0.04 | 0.71 | 0.30 | 85.29 | 34.03 |
| ASPL | 0.82 | 84.12 | 33.62 | 0.33 | 0.48 | 0.30 | 87.25 | 46.74 |
| ACE | 0.73 | 74.70 | 31.46 | **1.00** | N/A | 0.23 | 76.13 | 40.54 |
| ACE+ | **0.69** | **67.47** | **35.68** | 0.07 | 0.58 | 0.29 | 76.07 | 48.53 |
| ACE* | 0.76 | 72.52 | 32.76 | 0.75 | 0.47 | **0.13** | **73.92** | **76.50** |

indicating that our strategy of target selection is successful and robust with the motif in the target varying. Among all baselines, ASPL has the strongest performance. As mentioned in Section 3.3, this is the credit of introducing fine-tuning in the protection.

In addition to quantitative evaluation, we visualize the customization output images under different protection. Figure 1 showcases output images of LoRA and SDEdit under the protection of baselines and ACE/ACE+. We visualize the output images of ACE* as an ablation study in Figure 11. Notably, all baseline methods only add chaotic patterns to some parts of the output image. Diffusion customization still succeeds in generating images with correct contents and styles. In contrast, ACE/ACE+ covers the whole output image with high contrast and overly busy patterns, making the image completely unusable. This advantage distinguishes ACE/ACE+ from existing protection against unauthorized diffusion customization. More visualization is given in Appendix F.

## 4.3 User Study

Protections against unauthorized diffusion customization aim at making the output image of unauthorized customization unusable in graphic art industries. The gold standard for evaluating protection is user studies by graphic artists. Hence, we conduct a survey in an artist community on the comparative effectiveness of two most competitive protections: ASPL (Van Le et al., 2023), and ACE. Basically, our survey asks participants to pick the image with the worst quality out of two customization images under the protection of ASPL, and ACE, respectively. The strongest protection should make the output image of the worst quality. We omit the details to Appendix E. The survey covers 1304 artists, showing that ACE produces the worst customization image among 55% of the examples.

## 4.4 Transferability

Protections against unauthorized diffusion customization are based on white-box and model-dependent adversarial attacks. Hence, it is important to investigate the transferability of ACE/ACE+ over different diffusion models to validate their utility. We follow existing literature (Liang et al., 2023; Xue et al., 2023) to evaluate this transferability on three text-guided diffusion models: Stable Diffusion 1.4, Stable Diffusion 1.5, and Stable Diffusion 2.1. As mentioned in Section 2 and Section 4.1 and shown in Appendix E, we only consider these SD-family models because they cover most of unauthorized diffusion customization.

In this experiment, *backbone* means the model used to generate adversarial examples and *victim* means the model used in few-shot generation. We pick 100 images from CelebA-HQ, 20 in each of 5 groups, to generate adversarial examples on three backbone models and run SDEdit and LoRA with these adversarial examples on three victim models. The strength of SDEdit is set to 0.4 for visualizing the differences. Other experimental setups stay the same with Section 4.1. Table 2 shows that ACE/ACE+ can degrade diffusion customization even when the victim is different from the backbone. We omit the detailed explanation of the result to Appendix F.2.

Table 2: Transferability results of ACE (top) and ACE+ (middle) and visualization of ACE's output images (bottom). MS, CS, and CI stand for MS-SSIM, CLIP-SIM, and CLIP-IQA. Our methods maintain effective across different models by bringing different degradation to the output images.

| VICTIM | SD1.4 | | | SD1.5 | | | SD2.1 | | |
|---|---|---|---|---|---|---|---|---|---|
| | SDEdit | | LoRA | SDEdit | | LoRA | SDEdit | | LoRA |
| BACKBONE | MS ↓ | CS ↓ | CI ↑ | MS ↓ | CS ↓ | CI ↑ | MS ↓ | CS ↓ | CI ↑ |
| NO ATTACK | 0.85 | 91.71 | 20.32 | 0.85 | 91.16 | 19.22 | 0.80 | 79.00 | 16.78 |
| SD1.4 | 0.73 | 77.24 | 38.13 | 0.73 | 77.58 | 35.98 | 0.62 | 60.82 | 35.45 |
| SD1.5 | 0.73 | 77.29 | 36.65 | 0.73 | 77.50 | 32.11 | 0.72 | 60.10 | 45.05 |
| SD2.1 | 0.72 | 76.20 | 46.08 | 0.62 | 76.80 | 39.08 | 0.60 | 59.12 | 43.89 |
| VICTIM | SD1.4 | | | SD1.5 | | | SD2.1 | | |
| | SDEdit | | LoRA | SDEdit | | LoRA | SDEdit | | LoRA |
| BACKBONE | MS ↓ | CS ↓ | CI ↑ | MS ↓ | CS ↓ | CI ↑ | MS ↓ | CS ↓ | CI ↑ |
| NO ATTACK | 0.85 | 91.71 | 20.32 | 0.85 | 91.16 | 19.22 | 0.80 | 79.00 | 16.78 |
| SD1.4 | 0.67 | 66.83 | 40.69 | 0.67 | 66.40 | 31.53 | 0.58 | 56.41 | 67.96 |
| SD1.5 | 0.67 | 66.58 | 41.16 | 0.67 | 66.13 | 36.05 | 0.58 | 57.17 | 68.50 |
| SD2.1 | 0.67 | 66.33 | 41.80 | 0.67 | 57.17 | 41.96 | 0.58 | 57.27 | 73.59 |
| VICTIM | SD1.4 | | | SD1.5 | | | SD2.1 | | |
| BACKBONE | SDEdit | LoRA | | SDEdit | LoRA | | SDEdit | LoRA | |

## 4.5 ROBUSTNESS TO PURIFICATION

We investigate how ACE survives common denoising adversarial purification. We follow the setup in Liang et al. (2023) to consider Gaussian (Zantedeschi et al., 2017), JPEG (Das et al., 2018), Resizing (Xie et al., 2017), SR (Mustafa et al., 2019), and DiffPure (Nie et al., 2022). The adversarial budget is set as 8/255 while other setups follow Section 4.1. Among these purification methods, Gaussian adds Gaussian noise with variance 4 and 8 to the image. Two JPEG qualities, 20 and 70, are tried. Resizing includes two setups: 2x up-scaling + recovering (2x) and 0.5x down-scaling + recovering (0.5x). Other details are omitted to Appendix B.2.

Table 3 posts the CLIP-IQA for LoRA and MS-SSIM for SDEdit for ACE under denoising-based purification. Gaussian is the most useful method in purifying ACE, while other methods are disable in countering ACE. Notably, as the state-of-the-art purfication method, DiffPure tends to remove too many details of the image so that the customization output image loses some basic semantics of the reference image. Although our protections could be removed by this powerful purifier, it also greatly degrades the performance of diffusion customization, where we achieve our goal in a different way.

Table 3: Quantitative results and visualization of ACE under different purification methods. Most of purifications even degrade the image quality because they also erase semantic information of the reference image when purifying out protections.

| DEFENSE | NA | GAUSSIAN | | JPEG | | RESIZING | | SR | DIFFPURE |
|---|---|---|---|---|---|---|---|---|---|
| PARAM | | $\sigma = 4$ | $\sigma = 8$ | $Q = 20$ | $Q = 70$ | 2x | 0.5x | | |
| CLIP-IQA $\uparrow$ (LoRA) | 25.51 | 17.24 | 28.39 | 38.13 | 29.01 | 26.10 | 32.91 | 39.63 | 49.25 |
| MS-SSIM $\downarrow$ (SDEDIT) | 0.54 | 0.70 | 0.64 | 0.75 | 0.75 | 0.79 | 0.76 | 0.45 | 0.40 |

| DEFENSE | NA | GAUSSIAN | | JPEG | | RESIZING | | SR | DIFFPURE |
|---|---|---|---|---|---|---|---|---|---|
| PARAMS | | $\sigma = 4$ | $\sigma = 8$ | $Q = 20$ | $Q = 70$ | 2x | 0.5x | | |

Figure 3: Comparison between $\epsilon_{adv}$ and $\mathcal{B}_{spl}$ of ASPL and ACE. Blue-framed images are protected images that we use to compute $\epsilon_{adv}$. Red-framed images are clean images that we use to compute $\mathcal{B}_{spl}$. We visualize $\epsilon_{adv}, \mathcal{B}_{spl} \in \mathbb{R}^{64 \times 64 \times 4}$ as images with 4 channels. Complementary colors mean two pixel are reverse to each other. There is visible pattern correlation between $\epsilon_{adv}$ and $\mathcal{B}_{spl}$ in ACE.

## 5 DISCUSSION: HOW CAN TARGETED ATTACK BEATS UNTARGETED ATTACK?

While our methods empirically beat baselines in protecting images from unauthorized diffusion customization, it is non-trivial to explain our success. In this section, we discuss the process of fine-tuning diffusion models on protected images and provide one view of analysis on how these attack-based protections work and why ACE/ACE+/ACE$^*$ perform distinguishably better than untargeted baselines on countering LoRA fine-tuning.

First, we introduce the notation of our analysis. We use $x$ and $x'$ to denote the clean and protected image and $\theta$ and $\theta'$ for the original diffusion model (without customization) and the diffusion model customized on protected image $x$. Then, we introduce two intermediates that help us analyze the mechanism of attack-based protections:

1) $\epsilon_{adv}$ is the score function error of the protected image $x'$ in the original diffusion model $\theta$.

Figure 4: Demonstration of three steps in **Hypothesis 5.1**. First, Attacking step increases $\epsilon_{adv}$ of protected images. Second, Finetuning step trains the diffusion model to $\epsilon_{adv}$ by a bias $\mathcal{B}_{spl}$, whose direction is reversal to $\epsilon_{adv}$. Third, customized diffusion models include $\mathcal{B}_{spl}$ in sampling so that their output images appear to have chaotic patterns. This hypothesis explains why $\epsilon_{adv}$ and $\mathcal{B}_{spl}$ of ACE are reverse to each other as shown in Figure 3.

2) $\mathcal{B}_{spl}$ is the score-function error of clean image $x$ in the fine-tuned diffusion model $\theta'$.

$$\begin{cases} \epsilon_{adv} := \mathbb{E}_t \mathbb{E}_{z'(t)|z'(0)} \|s_\theta(z'(t), t) - \nabla_{z'(t)} \log p_t(z'(t)|z'(0))\|_2^2 \\ \mathcal{B}_{spl} := \mathbb{E}_t \mathbb{E}_{z(t)|z(0)} \|s_{\theta'}(z(t), t) - \nabla_{z(t)} \log p_t(z(t)|z(0))\|_2^2 \end{cases}$$

Intuitively, $\epsilon_{adv}$ reflects the difference between protected images and clean images from the perspective of the original diffusion model. $\mathcal{B}_{spl}$ reflects the error made by the customized diffusion model when predicting the score function of clean image.

To explain what happens when fine-tuning diffusion models on protected reference images, we visualize $\epsilon_{adv}$ and $\mathcal{B}_{spl}$ in ASPL and ACE, whose result is shown by Figure 3. Surprisingly, there is obvious reversal relationship between $\epsilon_{adv}$ and $\mathcal{B}_{spl}$ in ACE. The cosine similarity between $\epsilon_{adv}$ and $\mathcal{B}_{spl}$ in ACE is -0.3044. This indicates that what customized diffusion models learn from protected reference images is to output with a bias which reverses the score-function error of protected reference images. Base on this observation, we present a hypothesis on the mechanism of adversarial attacks on diffusion models, demonstrated by Figure 4

**Hypothesis 5.1** (Mechanism of attack-based protection on diffusion models (informal)). *First, attacks maximize the score-function error of protected images and introduce chaotic patterns to the error. Second, customized diffusion models are fine-tuned to minimize the score-function error of protected images, where a reversal bias is learned to overcome chaotic patterns in the error. Third, customized diffusion models also include the above reversal bias when generating customization images. This makes the customization output images come with reverse chaotic patterns.*

Following this hypothesis, we can give an explanation to the success of ACE: ACE uses a unified target to guide the chaotic pattern in all protected images. Therefore, the customized diffusion model learns to minimize one fixed chaotic pattern in score-function error when fine-tuning on different protected images. This induces the model to learn a fixed bias to reverse the fixed chaotic pattern. This bias is then brought to the output image generated by the customized diffusion model, explaining why we see a unified chaotic pattern in the customization output image under ACE.

In contrast, untargeted attacks do not unify the chaotic pattern in different protected images. The customized diffusion model cannot minimize score-function errors of different protected images by learning to reverse one chaotic pattern. Hence, we cannot see unified chaotic pattern in the customization output image. Although untargeted attacks could produce stronger chaotic patterns specifically for different protected images, these patterns do not work jointly or even conflict to each other in the training of customized diffusion model. This is the potential reason that targeted attacks beat untargeted attacks in this special task.

## 6 CONCLUSION

In this paper, we propose a simple yet effective protection method against unauthorized diffusion customization. Based on a targeted adversarial attack on diffusion models, we show that our method beats all existing protection through extensive experiments, with sound transferability and robustness to purification. We further analyze the mechanism of attack-based protections, providing a novel view to comprehend and explain the mechanism of adversarial attacks on diffusion models.

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

## A  Implementation and Experimental Details

**LoRA+DreamBooth** For the evaluation, we finetune LDM with on CelebA-HQ dataset using the prompt *a photo of a sks person*, which was first used in the paper of ASPL (Van Le et al., 2023). This is because CelebA-HQ consists of portraits of certain people. Similarly, we use the prompt *a photo of a sks painting* on the WikiArt dataset, because WikiArt consists of paintings by certain artists. The number of finetuning epochs is set to be 1000 which ensures LoRA on clean images achieves considerate performance.

**SDEdit** The strength of SDEdit is set to be 0.3, which makes sense because a higher strength modifies the input image too much and a lower one keeps too many details of the input image. We use a null prompt to avoid the effect of prompts in the generation.

**Metrics** For MS-SSIM, we use the default setting in the implementation [2]. CLIP-SIM computes the cosine similarity between the input images and the output images in the semantic space of CLIP (Radford et al., 2021) and is given by the following definition:

$$\text{CLIP-SIM}(X, Y) = 100 \times \cos(\mathcal{E}_{clip}(X), \mathcal{E}_{clip}(Y)) \tag{6}$$

where $\mathcal{E}_{clip}$ is the vision encoder of the CLIP Radford et al. (2021) model. CLIP-IQA is a non-reference image quality assessment metric that computes the text-image similarity between the image and some positive & negative prompts. In the official implementation [3], the author exploits prompts such as *Good image*, *Bad image*, and *Sharp image*. An image with high quality is expected to have high text-image similarity with positive prompts and low text-image similarity with negative prompts. In our experiments, we use the positive prompt *A good photo of a person* and the negative prompt *A bad photo of a person* for CelebA-HQ and the positive prompt *A good photo of a painting* and the negative prompt *A bad photo of a painting* for WikiArt. Since we want to measure how poor the output image quality is, we use the text-image similarity between output images and the negative prompt. A strong adversarial attack results in low quality of output images and a high similarity between output images and the negative prompt.

**Image Resolution** The standard resolution for SD1.x is 512, while the one for SD2.x is 768. For cross-model transferability experiments, we set the resolution of every model to 512, disregarding that the standard resolution of SD2.1 is 768. The reason for this uniform resolution is to avoid the resizing, which may introduce distortion to the attacks. However, as LoRA on SD2.1 naturally generate image of resolution 768, we still test LoRA performance on SD2.1 on resolution 768.

**Baseline Selection** Our works focuses on improving the protection effectiveness against unauthorized diffusion customization. Hence, we include baseline protection methods that 1) focus on effectiveness and 2) has open-sourced implementation or guidelines for reproduction. Also, to align with most baselines and make sure the comparison is fair, our evaluation is conducted on the same adversarial budget 3) under l2-constraint. Based on these three conditions, we select our baselines (Salman et al., 2023; Liang et al., 2023; Van Le et al., 2023). We notice that there are more existing works that contribute to the protection. However, some of them do not put their main contribution on improving effectiveness. For example, SDS (Xue et al., 2023) focuses on improving computational efficiency and MetaCloak (Liu et al., 2024b) focuses on robustness. Also, some existing works are lack of official open-sourced implementation (Liu et al., 2024a), while some others do not use l2-constraint (Shan et al., 2023a). Therefore, we do not include them in our baselines.

**Baseline Implementation** We use the official implementation of PhotoGuard [4], PhotoGuard+ [5], AdvDM [6], and ASPL [7] in our experiments. For PhotoGuard and PhotoGuard+, we follow the default setup in the official Python implementation file [8] [9], except for tuning the adversarial budget to $4/255$. We also set the surrogate model of PhotoGuard to be Stable Diffusion v1.5. PhotoGuard+ aims to

---

[2] https://github.com/VainF/pytorch-msssim
[3] https://github.com/IceClear/CLIP-IQA
[4] https://github.com/MadryLab/photoguard
[5] https://github.com/MadryLab/photoguard
[6] https://github.com/mist-project/mist
[7] https://github.com/VinAIResearch/Anti-DreamBooth
[8] https://github.com/MadryLab/photoguard/blob/main/notebooks/demo_simple_attack_inpainting.ipynb
[9] https://github.com/MadryLab/photoguard/blob/main/notebooks/demo_complex_attack_inpainting.ipynb

attack an impainting task, and we simply set the impainting mask to 1 for all pixels, as both SDEdit and LoRA operates on full pixels. For AdvDM, we also follow the default setup in the official implementation. For ASPL, we directly use the official implementation except for transferring it to LoRA. The default setup sets 10 steps of training LoRA and 10 steps of PGD attacks in every epoch. However, the default epochs of ASPL is too time-consuming. Therefore, we tune the total epochs of one single attack to be 4, which is a fair comparison for our method.

**Hyperparameters & Implementation Details** For ACE+, the loss weight $\alpha$ is set to be $10^2$ empirically. LoRA is done for 5 iterations while each iteration finetunes 10 steps. The learning rate is $10^{-5}$. To keep consistent to the real application, we use bfloat16 accuracy for ACE/ACE+. All experiments except PhotoGuard+ are done on one NVIDIA RTX 4090 GPU. The implementation of PhotoGuard+ is done on one NVIDIA RTX A100 GPU.

**Visualization of Variables in the Latent Space of LDM** We describe how to visualize variables in the latent space of LDM, including $\epsilon_{adv}$, $\mathcal{B}_x$, and $\mathcal{B}_{spl}$. These variables are of size $64 \times 64 \times 4$. Given a variable $E \in \mathbb{R}^{64 \times 64 \times 4}$, we first do normalization by $E' = \frac{E - min(E)}{max(E) - min(E)}$. After the normalization, we directly plot $E'$ as a heatmap. The used color bar is demonstrate in Figure 5.

# B    ADDITIONAL EXPERIMENTS

## B.1    COMPUTATION COST

Computation cost is crucial to the utility of adversarial attacks on LDM for. First, they are designed for the sake of private image protection involved in the daily routine of human. This asks for the time efficiency of the attack. Second, the main body of the users are non-developers who do not have access to GPUs with large VRAM. Hence, the attack needs to be memory-efficient.



### B.1.1    ANALYSIS OF THE COMPUTATION COST IN ACE/ACE+

Figure 5: An example for visualization. We use the same color bar in all visualizations.

**Time Cost** As demonstrated in Algorithm 1, ACE/ACE+ consists of two computing processes: parameter updating (model training, step 5-7 in Algorithm 1) and input updating (PGD, step 8-12 in Algorithm 1). As stated in Appendix A, we set the iteration number to be 4, the parameter updating step number for each iteration to be 10, and the input updating step number to be 50 for each image. Note that the number of image is 20 in our experimental setup. Therefore, we run $5 \times 10 = 50$ steps of parameter updating and $4 \times 50 \times 20 = 4000$ steps of input updating. The step number ratio is 1:80. Hence, although one single parameter updating step may consume more time cost than one input updating step[10], the main proportion of time cost is still cost by input updating. This conclusion is validated by experiments in the following.

**Memory Cost** The memory cost of ACE/ACE+ consists of three occupies, which store model weights, the computational graph, and the optimizer states, respectively. In the adversarial attack, we only optimize the inputs so the memory used to store optimizer states is small. We save memory cost mainly by decreasing the memory to store model weights and computational graph. We introduce two optimizations to for reducing memory cost, whose efficiency is validated by experiments in the following.

### B.1.2    COMPUTATION COST OPTIMIZATIONS IN ACE/ACE+

To reduce and balance the computation cost of ACE/ACE+, we introduce the following two optimizations:

**xFormers** We leverage xFormers (Lefaudeux et al., 2022) to reduce the memory cost used to store the computational graph. xFormers is a toolbox that provides memory-efficient computation operators

---

[10]As advised by Zhang et al. (2019), the time cost bottleneck is the back-propagation operation, which is included by both parameter updating steps and input updating steps. Hence, their single step time cost should be similar.

for training and inference of transformer-based modules. We use their attention operator in the computation of cross attention layers of UNet in LDM when computing gradients.

**Gradient Checkpointing** Chen et al. (2016) A common tool of memory-efficient training is Gradient Checkpointing. Gradient Checkpointing separates a neural network into blocks. In forward-propagation, it only stores the activation. The back-propagation is done block by block. For each block, it reconstructs the forward computational graph within the block with the stored activation. Then, it constructs the backward computational graph within the block and compute the gradient over the activation. This greatly reduces the GPU memory at the cost of computing time. To balance the memory and time cost, we only apply gradient checkpointing in the down-block, mid-block, and up-block of the UNet.

### B.1.3 EVALUATION: MEMORY COST

Table 4 shows the computation cost of ACE and existing attacks. ACE enjoys the lowest memory cost, 5.77 GB, compared to baselines. For the time cost, ACE takes 70.49 seconds to process one image, which is acceptable in real-world private image protection. We evaluate the memory cost

Table 4: Computation cost of our method and baselines

| Method | Memory/GB | Time/(seconds/image) |
|---|---|---|
| PhotoGuard | 6.16 | 21.06 |
| PhotoGuard+ | 16.79 | 878.43 |
| AdvDM | 6.28 | 13.45 |
| ASPL | 7.33 | 55.63 |
| ACE&ACE+ | 5.77 | 70.49 |

of baselines and that of ACE/ACE+. We apply baselines and ACE/ACE+ on a group of 20 images from CelebA-HQ and record the maximum memory occupation of the runtime. To better simulate the real-world scenario of artwork protection, this experiment is run on an NVIDIA RTX 3080Ti[11], which is an off-the-shelf consumer-level GPU. Other setups stay the same as the setup in Section 4.1.

Table 4 shows that the memory cost of ACE/ACE+ is 5.77 GB, lower than those of all baselines. We highlight that our GPU memory cost is lower than 6GB. This means that **ACE/ACE+ is able to run on most of the consumer-level GPUs**, which has not been achieved by any existing methods. This helps popularize the application of adversarial attacks on LDM as a practical tool for artwork protection. Additionally, we check the memory occupation of two kinds of updating steps. Parameter updating step takes 4.60 GB and input updating step 5.77 GB. This result proves that input updating is the bottleneck for GPU memory cost in ACE/ACE+.

### B.1.4 EVALUATION: TIME COST

We evaluate the time cost of baselines and that of ACE/ACE+. We apply baselines and ACE/ACE+ on a group of 20 images from CelebA-HQ and record the time cost of the runtime. To better simulate the real-world scenario of artwork protection, this experiment is run on an NVIDIA RTX 3080Ti, which is an off-the-shelf consumer-level GPU[12]. We rerun the experiment for three times to reduce the effect of randomness. Other setups stay the same as the setup in Section 4.1.

Table 4 shows that the **time cost to process one image** of ACE/ACE+ is 88.11 seconds, 27% longer than that of ASPL (Van Le et al., 2023), our strongest baseline. Considering that one human artist does not need to process dozens of artworks normally, the time cost of ACE/ACE+ is acceptable in real-world application.

We also investigate the time cost proportion between two kinds of updating steps. We rerun ACE for three times following the above setup and record the **total time cost to process 20 images** of

---

[11]PhotoGuard+ is an exception because it requires over 16 GB GPU memory. Hence, we run it on an NVIDIA A100 80GB.

[12]Similar to the case of the memory cost experiment, PhotoGuard+ is evaluated on A100 80GB. Since PhotoGuard+ is very inefficient, we only post its computation cost as a reference.

all parameter updating steps and input updating steps separately. As shown in Table 5, parameter updating steps only take $1.46\%$ of the whole time cost, while input updating steps take the rest $98.54\%$. This result validates that input updating takes the larger proportion of time cost.

Table 5: Time cost of parameter updating and input updating in ACE (processing 20 images).

|  | Parameter Updating/seconds | Input Updating/seconds | Total/seconds |
|---|---|---|---|
| Run 1 | 20.61 | 1383.85 | 1405.46 |
| Run 2 | 21.24 | 1396.41 | 1417.65 |
| Run 3 | 19.92 | 1386.30 | 1406.21 |
| Average | 20.59 | 1389.18 | 1409.78 |
| Proportion | 1.46% | 98.54% | 100% |

## B.2 ROBUSTNESS TO PURIFICATION METHODS

As discussed in Section 4.5, we conduct an experiment to investigate how ACE survives purification methods. We consider common denoising-based purification methods, including Gaussian (Zantedeschi et al., 2017), JPEG (Das et al., 2018), Resizing (Xie et al., 2017), SR (Mustafa et al., 2019), following the setup in existing works (Liang et al., 2023). This is because other purification methods are either not compatible or not available in the context of LDM.

We use these purification methods to denoise the adversarial perturbations on the protected images. After that, we apply SDEdit and LoRA on these purified images. The experimental setup follows Section 4.1 except for the adversarial perturbation constraint, which is set to be 8/255. This is still a small constraint compared to the setup of existing protections (For comparison, Salman et al. (2023) and Van Le et al. (2023) use 16/255). The purification methods include Gaussian (Zantedeschi et al., 2017), JPEG (Das et al., 2018), Resizing (Xie et al., 2017), SR (Mustafa et al., 2019). Specifically, Gaussian adds Gaussian noise of standard variance 4 and 8 to the protected image. We try two JPEG compression qualities, 20 and 70. For Resizing we have two setups, 2x up-scaling + recovering (denoted by 2x) and 0.5x down-scaling + recovering (denoted by 0.5x). The interpolation algorithm in resizing is bicubic.

The quantitative result is demonstrated in Table 3. It shows that ACE and ACE+ still have strong impact on the output image quality of few-shot generation after processing by purification. The CLIP-IQA score of most cases even increase after the purification. Hence, we believe that our adversarial attack has enough robustness to purification methods.

Qualitatively, Table 3 also visualizes the output images of SDEdit and LoRA referring to the protected images processed by different purification methods. For both method, they still have strong performance under most of the cases except for Gaussian Noise($\sigma = 8$) and JPEG compression (quality= 20). However, in the exception cases, the defense has added visible degradation to the image, which also heavily affect both LoRA and SDEdit process. For example, LoRA learns to produce images comprised of small squares due to a hard compression of quality 20. And SDEdit produces images of visible Gaussian noise when adding Gaussian noise of $\sigma = 8$. It's noteworthy that both ACE and ACE+ seems to be strengthened after SR purification, which is an intriguing phenomena.

## B.3 PERFORMANCE ON DREAMBOOTH

Although LoRA+DreamBooth is the most widely-used customization method, we also follow ASPL (Van Le et al., 2023) to conduct experiments on DreamBooth, as robustness evaluation of our method. We compare ACE with ASPL, the strongest baseline, on DreamBooth. Except for replacing the finetuning method by DreamBooth, all experimental setups stay the same as Section 4.1 and Appendix A. We visualize the output images of DreamBooth under ACE and ASPL in Figure 6, respectively. ASPL mainly hurts the diversity of output images while ACE severely degrades the image quality and makes the output images unusable by adding perceptible chaotic texture.

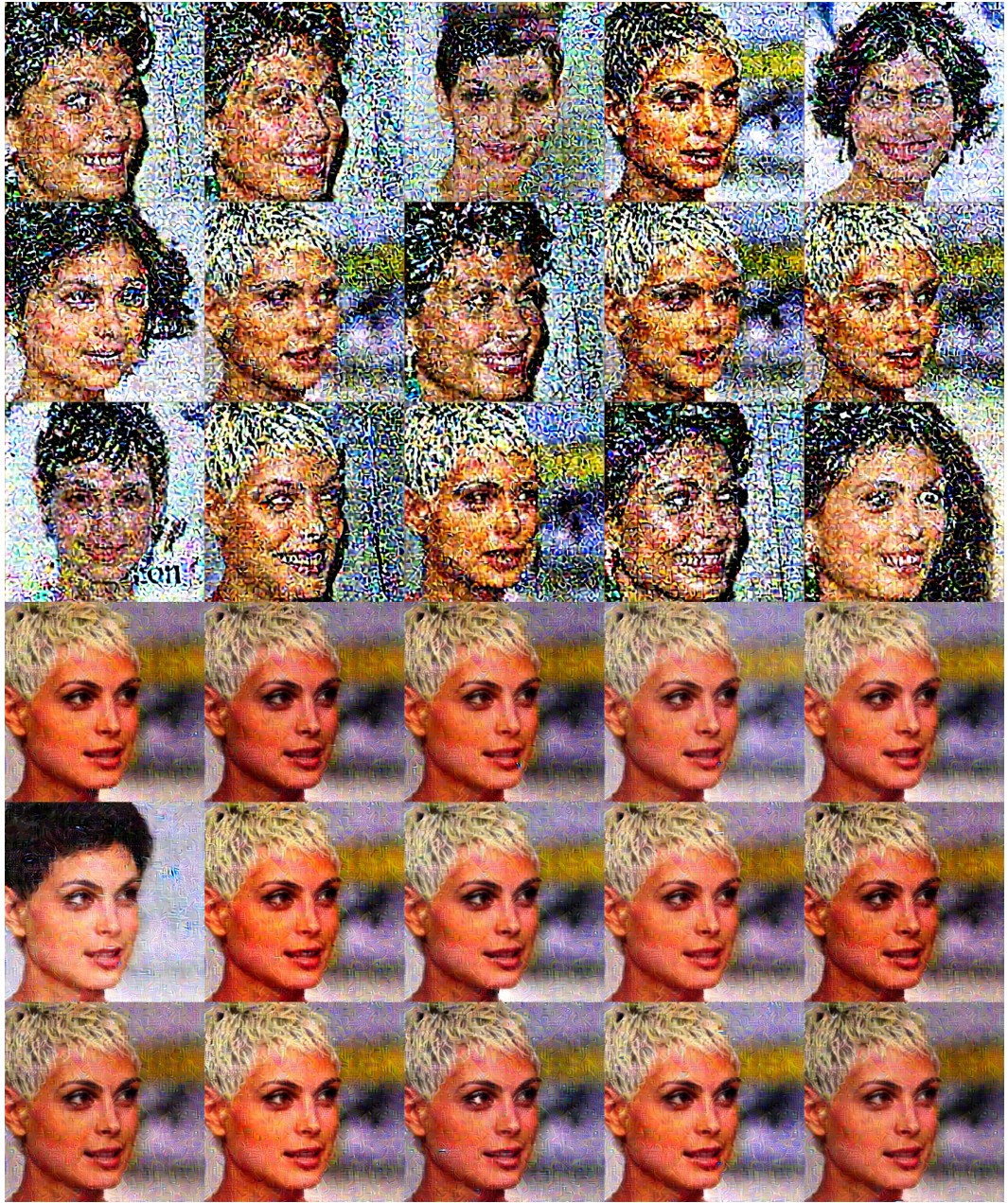

Figure 6: Comparison of ACE and ASPL on DreamBooth. The top three rows: output images of DreamBooth under ACE. The bottem three rows: output images of DreamBooth under ASPL. The adversarial budget is $\zeta = 4/255$. ASPL mainly hurts the diversity of output images while ACE directly degrades the image quality.

### B.4 PERFORMANCE ON STABLE DIFFUSION 3

To investigate whether ACE works on diffusion models based on Diffusion Transformers (DiTs) (Peebles & Xie, 2023) and Rectified Flow (Liu et al., 2022), we evaluate ACE on Stable Diffusion 3 (Esser et al., 2024), one state-of-the-art diffusion models for text-to-image generation. We follow our default setup on LoRA-DreamBooth, except for: 1) the image resolution is 1024 as the default of Stable Diffusion 3, 2) the adversarial budget is set to $8/255$, and 3) the training steps of LoRA are adapted to 2000 for training convergence. We train LoRA on both the clean and the protected subsets of CelebA-HQ. Then, we calculate the CLIP-IQA for both groups.

The CLIP-IQA of LoRA on the clean subset is $0.2650$, while that of LoRA on protected subset is $0.4963$. This validates that ACE is effective on Stable Diffusion 3. **ACE is the first protection method that proves to work on Stable Diffusion 3 or other DiT-based diffusion models.** Notably, ACE seems to work on Stable Diffusion 3 in a mechanism different from that on early versions of Stable Diffusion. Instead of degrading the quality of output images, ACE deviates the semantic of output images. For example, output images with LoRA trained on portraits of a person A will be portraits of other irrelevant persons. We visualize this phenomena in Figure 7. This means that one attack could perform in different ways on diffusion models with different architectures, which is a brand new finding about adversarial attacks on diffusion models. We leave its explanation to future work.

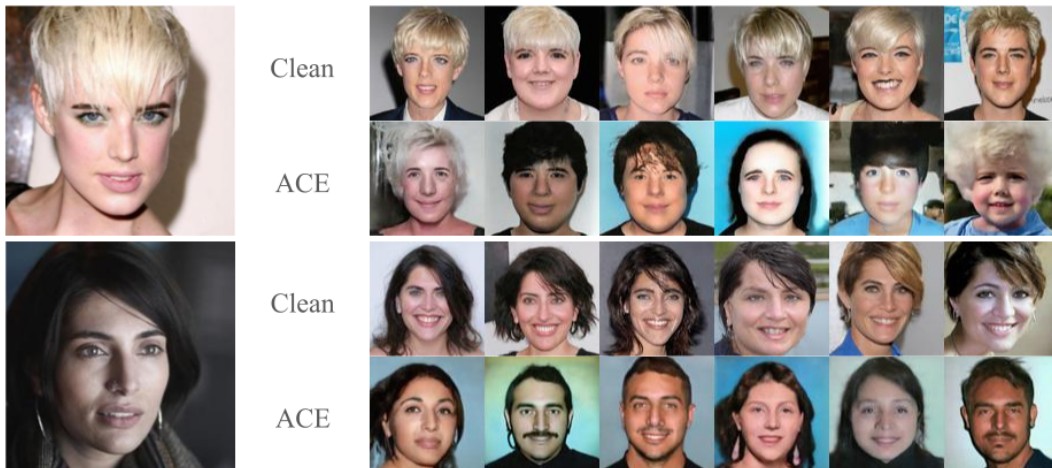

Figure 7: Comparison of output images from LoRA trained on clean images and protected images by ACE on Stable Diffusion 3. On Stable Diffusion 3, LoRA under ACE will deviate the semantics in the training images and generate irrelevant images, for example, generating portraits different from the portraits in the training images.

### B.5 ABLATION STUDIES

#### B.5.1 TARGETS AND OBJECTIVES

We conduct ablation studies on the effect of targets and objectives. For targets, we use two natural images, one human-face photo from CelebA-HQ and one artwork from WikiArt, as alternative images for the target $\mathcal{T}$. This is recommended by Van Le et al. (2023). For objectives, we use ASPL-T as the alternative objective. The experiment setup follows the LoRA setup in Section 4.1.

Table 6 demonstrates the sampling outputs under different targets and objectives. Under the small adversarial budget $\zeta = 4/255$, ASPL-T fails in degrading the quality of sampling outputs with all targets. By contrast, ACE adds obvious texture to the sampling outputs with different targets. Among them, our target shows distinguishing superiority in adding very chaotic texture to the sampling output. The result validates that both the objective of ACE in Section 3.1 and the target $\mathcal{T}$ selected in Section 3.2 contribute to the performance of our method.

Second, we investigate the impact of two vision variables in the target image $\mathcal{T}$: pattern density and contrast. The setup follows that in the first experiment above. The result is visualized in Figure 8.

Table 6: Ablation study on the effect of different targets and objectives.

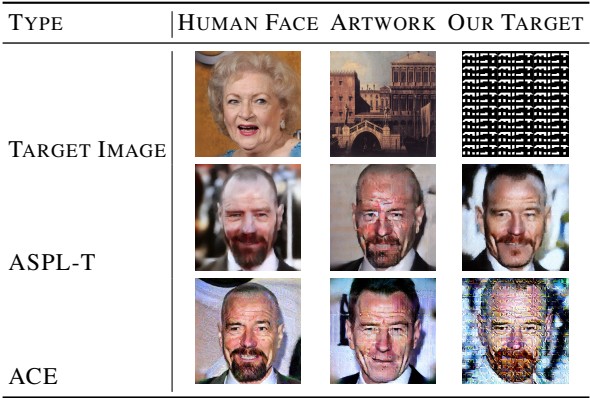

| TYPE | HUMAN FACE | ARTWORK | OUR TARGET |
|------|------------|---------|------------|
| TARGET IMAGE | | | |
| ASPL-T | | | |
| ACE | | | |

When the pattern is sparse, increasing pattern density yields better attacking performance. When the pattern is dense, increasing pattern density then leads to worse performance. Hence, in practice, we choose an appropriate pattern density. As for contrast, the performance of the attack increases as the pattern's contrast goes higher. We then let our target be with the highest contrast.

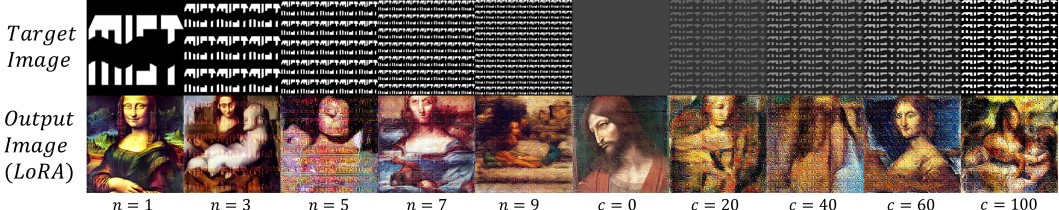

Figure 8: Target images with different pattern repetition and contrast results in different effects.

### B.5.2 PROMPTS IN LORA

In real life usage, malicious ones may use different prompts to finetune the LDM. Hence, we investigate whether ACE maintains effectiveness under different prompts in the finetuning. We select one 20-image group from CelebA-HQ and use ACE to generate protected images under adversarial budget $4/255$ and prompt *a photo of a sks person*. We then train LoRA with four different prompts:

- a photo of a sks person
- a photo of a pkp
- a photo of a pkp woman
- an image of a pkp woman

We visualize the output images in Table 7. The result shows minor degradation in the effectiveness of attack when ACE handles more and more unfamiliar prompts. However, there still exists strong visual distortion under all three unknown prompts.

### B.5.3 PERTURBATION MAGNITUDE

One interesting question is how the degradation of LDM's performance is related to the magnitude of adversarial perturbations. We conduct an experiment to provide a quantitative answer. We tune the adversarial budget $\zeta$ to be $\{2, 4, 6, 8, 12\}$ and keep the other setup consistent as that in Section 4.1. In addition to the budget, we also post PSNR to measure the magnitude of adversarial perturbation. For the degradation of LDM's performance, we post CLIP-IQA for LoRA and MS-SSIM & CLIP-SIM for SDEdit.

Figure 9 demonstrates the trend of performance degradation with the increase of magnitudes of adversarial perturbations. For SDEdit, the performance degradation always grow together with the

Table 7: LoRA output images with different prompts under the attack of ACE. Red text notes the difference between the prompt in generation and the prompt in finetuning.

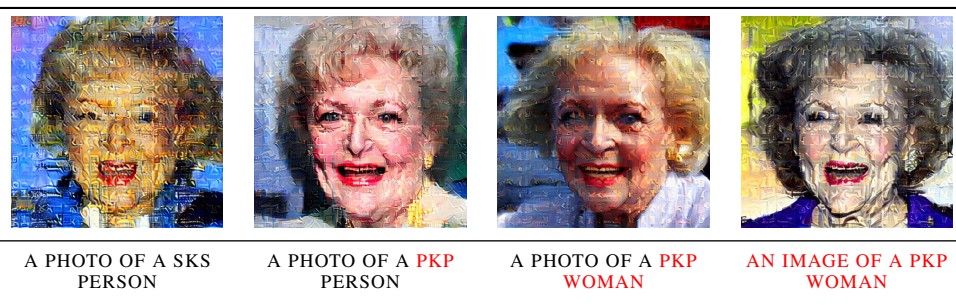

| A PHOTO OF A SKS PERSON | A PHOTO OF A PKP PERSON | A PHOTO OF A PKP WOMAN | AN IMAGE OF A PKP WOMAN |

perturbation magnitude, because SDEdit directly edits the protected image. For LoRA, performance degradation continues to grow when the magnitude increases from 2 to 8. When the magnitude grows from 8 to 14, however, the performance degradation holds still. This indicates that $\zeta = 8$ is approximately a threshold for the performance degradation in LoRA.

Nevertheless, the main concern of increasing adversarial budgets is the visibility of adversarial perturbations. According to our user survey among human artists, $\zeta = 4$ is an appropriate budget. This is the reason why we conduct experiments on $\zeta = 4$.

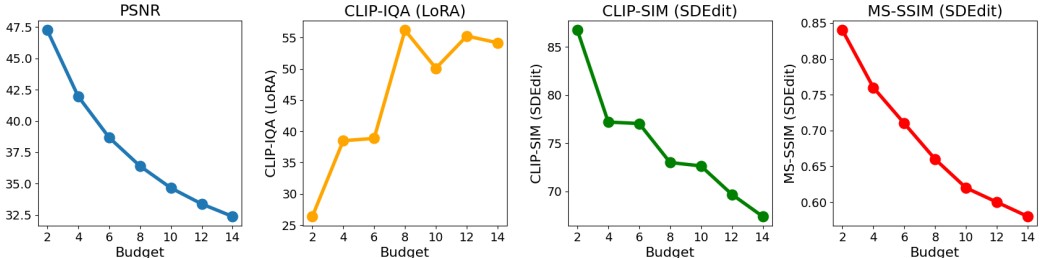

Figure 9: On the relation between perturbation magnitudes and degradation performance. The left-most figure depicts the PSNR between the protected images and the original images. This could be seen as another metric for adversarial perturbation budgets.

### B.5.4 ACE WITH A NEW TARGET

From Appendix B.5.1, we know that a good target for ACE should be with high contrast and repeated patterns. To further validate this insight, we design from scratch a new target (See Figure 10) and conduct full sets of our experiments accordingly. We pick the logo of NeurIPS and repeat it by $2 \times 8$ in an black image. We denote ACE with the new target by ACE*. All other setups follow Section 4.1.

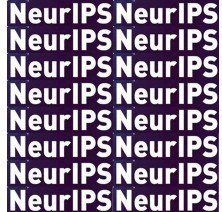

Figure 10: New target used by ACE*.

We compare ACE* with the strongest baseline ASPL (Van Le et al., 2023) and our ACE/ACE+ in Table 1, with visualization given by Figure 11. ACE* performs similarly or even better than baselines. This shows that ACE can yield even better performance by carefully picking the target following our insights, providing stronger evidence for our target selection.

## C LIMITATIONS & SOCIETAL IMPACTS

**Limitations** First, due to the quick progress of the few-show generation empowered by LDM, ACE may need updates to cover the novel artwork copying pipelines in the future. Second, this paper focuses on explaining the mechanism of adversarial attacks on LDM empirically, which leads to a more powerful method. Hence, we do not provide rigorous theoretical proofs for our explanation

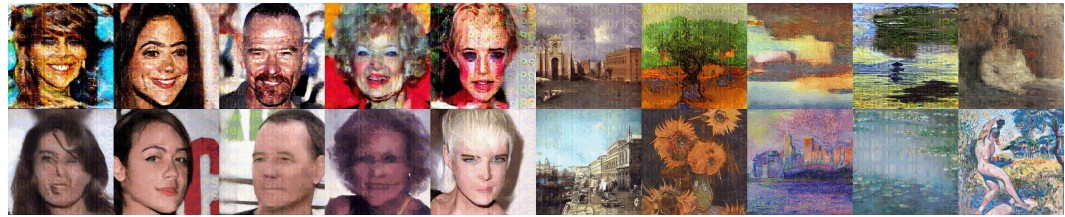

Figure 11: Visualization of LoRA (top) and SDEdit (bottom) output images under ACE*.

but focus on validating it with empirical observation. A rigorous theoretical framework for this mechanism is then left to be explored in the future work.

In addition, we follow current works to use l2-distance to constrain our adversarial perturbations, which makes comparisons fair. This design, however, **could be vulnerable** to some new advanced purification methods, for example, IMPRESS (Cao et al., 2023) and GrIDPure (Zhao et al., 2024). We believe this could be overcome by rectifying the constraint and designing patterns that hide our protection deeper in the original image. However, this is out of the scope of this paper which focuses on optimizing the attack objective. We believe our stronger objective can contribute to real-world protection in combination with a potential robust constraint and pattern, where we leave to the future work. Also, our robustness experiments show that our method has robustness against general purification, keeping its essential utility. Hence, we leave the improvement of the robustness against advanced purification to the future work.

**Societal Impacts** This paper presents work whose goal is to prevent the unethical abuse of the new generation AIGC. This is a crucial topic because of the legislative lag and the rapid progress of machine learning. The recent world has witnessed various cases concerning the copyright, privacy, and ethical problems of AIGC, most of which have not been solved by either technology or the law. Our work aims at providing stronger protection against unauthorized image copying and editing, which is only a small part of these problems. We thus believe that our work is completely ethical and sincerely hope that more attention will be paid to this field.

## D    KEY DIFFERENCES BETWEEN OUR METHOD AND OTHER TARGETED ATTACK

ACE/ACE+ are not the first targeted attack on LDM. ASPL-T is a general targeted form of attacks (Van Le et al., 2023). Here, we compare the objective function of ACE and ASPL-T [13]:

$$\textbf{ACE:}\quad \min_{x'} \mathbb{E}_t \mathbb{E}_{z'(t)|z'(0)} \|s_\theta(z'(t), t) - \boldsymbol{\mathcal{T}}\|_2^2$$

$$\textbf{ASPL-T:}\quad \max_{x'} \underbrace{\mathbb{E}_t \mathbb{E}_{z'(t)|z'(0)} \|s_\theta(z(t), t) - \nabla_{z(t)} \log p_t(z(t)|z(0))\|_2^2}_{\textbf{ASPL}} - \|z'_\theta(t-1) - z^{\boldsymbol{\mathcal{T}}}(t-1)\|_2^2$$

$$\text{where } z'_\theta(t-1) = \frac{1}{\sqrt{1-\beta_t}}(z'(t) + \beta_t s_\theta(z(t), t)) + \sqrt{\beta_t}\boldsymbol{\sigma_z}, \boldsymbol{\sigma_z} \sim \mathcal{N}(\boldsymbol{\sigma_z}; 0, \boldsymbol{I})$$

$$\text{and } z^{\boldsymbol{\mathcal{T}}}(t-1) = \sqrt{\beta_t}\boldsymbol{\mathcal{T}} + \sqrt{1-\beta_t}\boldsymbol{\sigma}, \boldsymbol{\sigma} \sim \mathcal{N}(\boldsymbol{\sigma}; 0, \boldsymbol{I})$$

$$(7)$$

ASPL-T adds a targeted guidance to ASPL. It is intuitive that it pushes the predicted previous latent representation $z'_\theta(t-1)$ to the ground truth previous latent representation $z^{\boldsymbol{\mathcal{T}}}(t-1)$ of a target $\boldsymbol{\mathcal{T}}$. This sub-objective is jointly optimized with the original objective of ASPL, which maximizes the training loss of LDM. However, according to the result posted in its original paper (Van Le et al., 2023), ASPL-T is deterior to ASPL in performance.

Compared to the intuitive approach of introducing the target, ACE considers the target directly as the only guidance for the predicted score function $s_\theta(z'(t), t)$. It is non-trivial because it does not correspond to an intuitive explanation within the framework of score-based generative modeling,

---

[13]The original paper does not provide the concrete objective function of ASPL-T so we refer to its implementation on GitHub: `https://github.com/VinAIResearch/Anti-DreamBooth/blob/main/attacks/aspl.py`

unlike ASPL-T. We can only design it with acknowledging that all attacks work by biasing the predicted score function $s_\theta(z'(t), t)$.

Glaze (Shan et al., 2023a) is another well-known targeted attack. Two core designs of our method are both different from Glaze:

**a) objective** ACE minimizes the distance between diffusion predicted latent noise and a fixed latent. We have both the VAE $\mathcal{E}$ and the UNet $\theta$ (or Transformers in DiT architecture) involved. By contrast, Glaze minimizes the feature distance between the protected image and the target image in the VAE representation space. Only the VAE $\mathcal{E}$ is used in this process.

$$
\begin{aligned}
&\textbf{ACE:} \quad \min_{x'} \mathbb{E}_t \mathbb{E}_{z'(t)|z'(0)} \| s_\theta(z'(t), t) - \mathcal{T} \|_2^2 \\
&\textbf{Glaze:} \quad \min_{x'} \| z'(0) - z_{\mathcal{T}}(0) \|_2^2 \\
&\qquad \text{where } z'(0) = \mathcal{E}(x'), z_{\mathcal{T}}(0) = \mathcal{E}(x_{\mathcal{T}}), \| x' - x \| < p
\end{aligned}
\tag{8}
$$

Here, $\| x' - x \|$ is some distance metric and instantiated as LPIPS (Zhang et al., 2018) in the implementation.

**b) target** ACE exploits the latent of a chaotic-pattern image $x_{\mathcal{T}}$ as the target. Glaze uses the style-transfered version of the original image as the target image.

$$
\begin{aligned}
&\textbf{Glaze:} \quad x_{\mathcal{T}} := \Omega(x, T) \\
&\qquad \text{where } \Omega \text{ is a style transfer pipeline of Stable Diffusion} \\
&\qquad \text{and } T \text{ is the target style other than the original style of the image } x
\end{aligned}
\tag{9}
$$

# E TOY SURVEY ON DIFFUSION CUSTOMIZATION

**User Study** To further compare ACE with the previous SOTA method ASPL and gain more feedback from the artist community, we conduct a survey on the effectiveness of ACE and ASPL. We choose ten 20-image subsets from our dataset and generate 100 customized images using LoRA-DreamBooth for each subset, under the protection of ACE and ASPL respectively. All hyperparameters are the same as mentioned in A. To test ACE's effectiveness against ASPL, we randomly sample one ACE-protected image and one ASPL-protected image for comparison. The artists are asked to vote for the lower quality one. We collect a total of 1451 valid votes and ACE has a win rate of 55.48%. This further showcases ACE's effectiveness in a real-world setting. To relieve the artists' workload, we do not include more baseline methods in the survey.

**Survey on Diffusion Customization Methods** Additionally, we investigate 50 highest-rated models in CivitAI [14], a popular platform for sharing diffusion customization models. Among them, 78% are LoRA models, which demonstrate the dominance of LoRA-DreamBooth in current diffusion customization applications.

# F VISUALIZATION

In this section, we visualize the comparison result between our proposed methods, ACE and ACE+, and baseline methods.

## F.1 PROTECTED IMAGES

Figure 12 and Figure 13 demonstrate protected images generated by our adversarial attack and ASPL, respectively. Protected images by both methods under adversarial budget $\zeta = 4/255$ show few differences between real images. However, the adversarial perturbations on ACE's protected images appear to be more smooth than those on ASPL's. This makes ACE more acceptable in real-world artwork protection.

---

[14]https://civitai.com/models

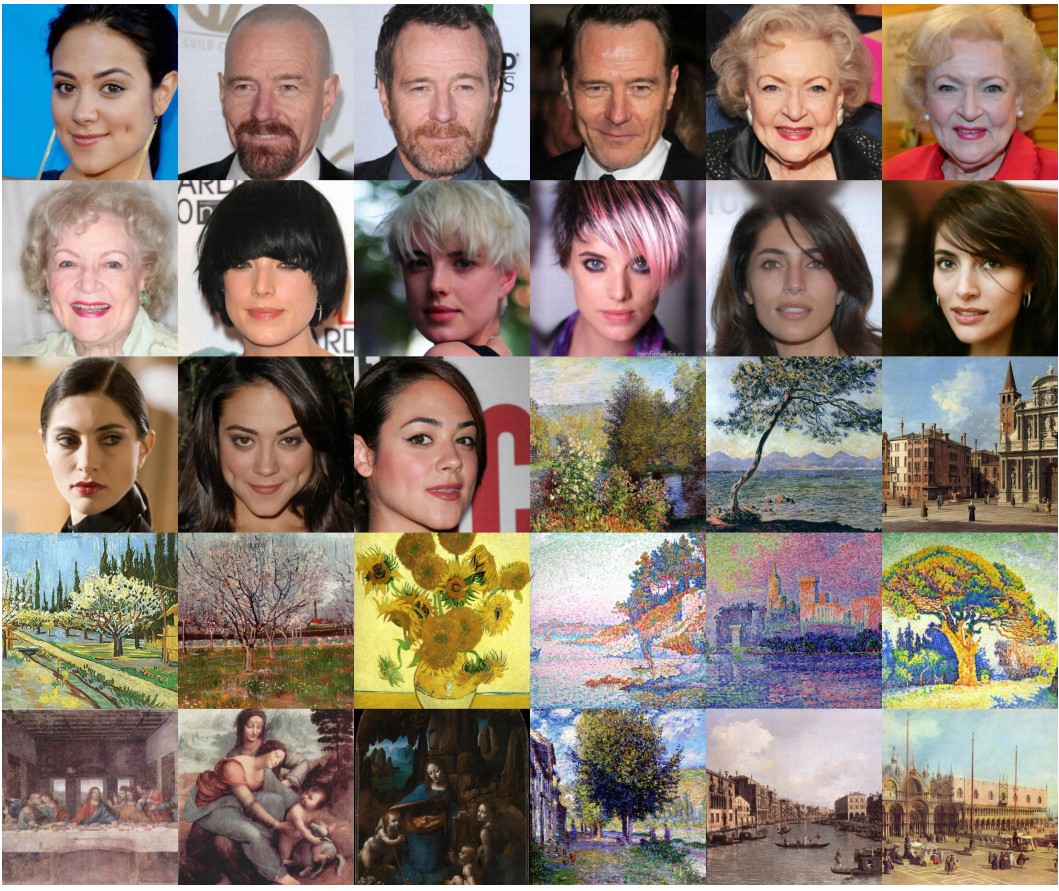

Figure 12: Protected images by ACE with the adversarial perturbation budget $\zeta = 4/255$. The perturbation is quite small and almost human-invisible, making the protected images resemble real images.

## F.2 TRANSFERABILITY

We visualize the output images of different victim models under ACE by different backbone models in Table 2. As shown in the table, our method shows a strong consistency among different models. An exception is that when using SD2.1 as the victim model, it tends to fail in LoRA training(not learning the right person) instead of learning the strong semantic distortion from the target image. However, the model does learn the right person when no attack is conducted. Also, the SDEdit process is extra strong when victim is SD2.1. We attribute this phenomena to the resolution mismatch. SD2.1 is trained to receive images of resolution 768, while we actually fed it with images of resolution 512. This may leads to different behaviour of SD2.1. Also, the resolution mismatch between SD1.x and SD2.x may be the main reason for the performance degradation when using SD1.x as victim and SD2.1 as backbone.

## F.3 OUTPUT IMAGES OF SDEDIT

Figure 14 visualizes the output images of SDEdit under different adversarial attacks. All adversarial attacks are budgeted by $\zeta = 4/255$. Two proposed methods add obvious noise to the output image, compared to no attack and three baseline methods, Photoguard, AdvDM, and ASPL. Note that the texture in the output images of ACE/ACE+ shows consistency with the pattern demonstrated in Figure 3.

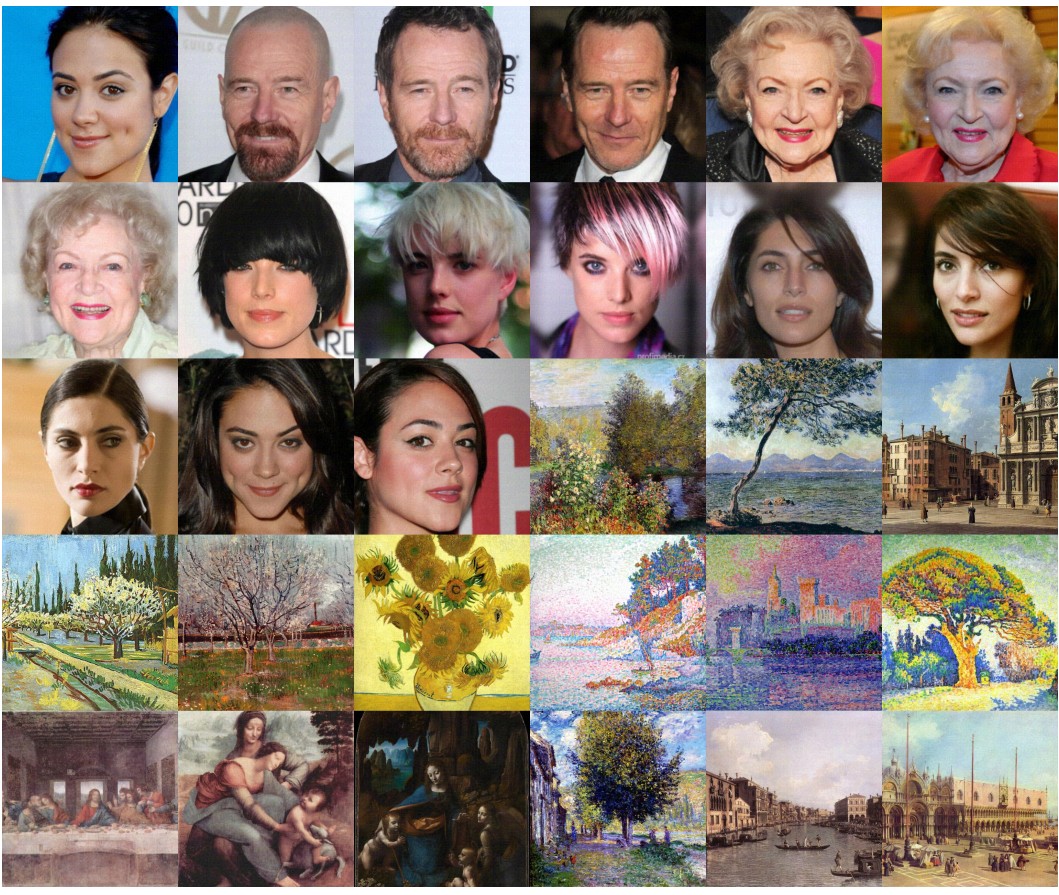

Figure 13: Protected images by ASPL with the adversarial perturbation budget $\zeta = 4/255$. Their adversarial perturbations are more visible than those by ACE.

## F.4    OUTPUT IMAGES OF LoRA

Figure 15, 16, 17, and 18 show the output images of LoRA under different adversarial attacks. All adversarial attacks are budgeted by $\zeta = 4/255$. Note that the texture in the output images of ACE/ACE+ shows consistency with the pattern of the sampling bias $\mathcal{B}_{spl}$ demonstrated in Figure 3. It is naturally because the texture is caused by the sampling error, which is accumulated by $\mathcal{B}_{spl}$.

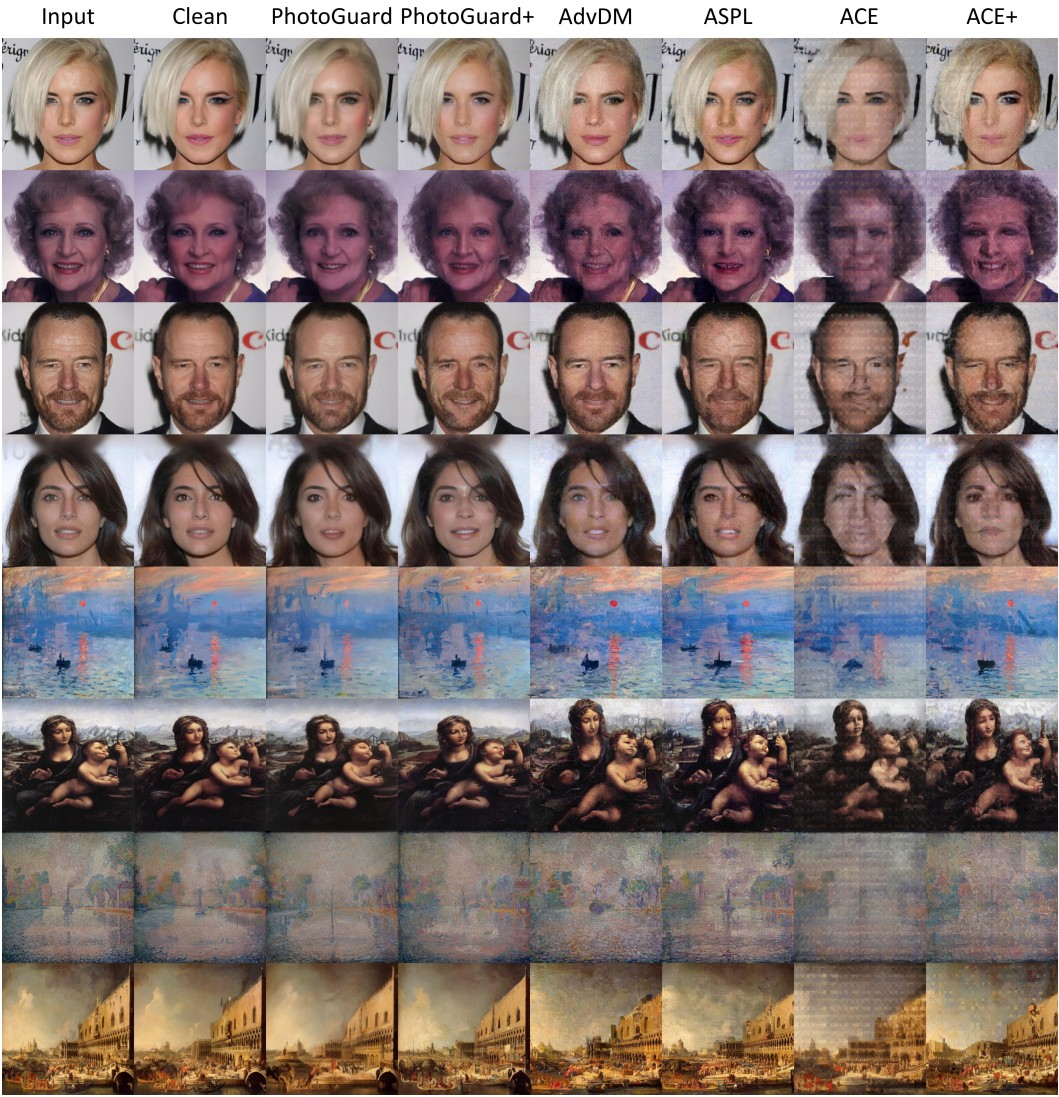

Figure 14: Output images of SDEdit under different adversarial attacks. With the same perturbation budget, our attacks better interfere the image quality compared to three baseline methods. Specifically, ACE adds chaotic texture to the image. ACE erases some contents of the image.

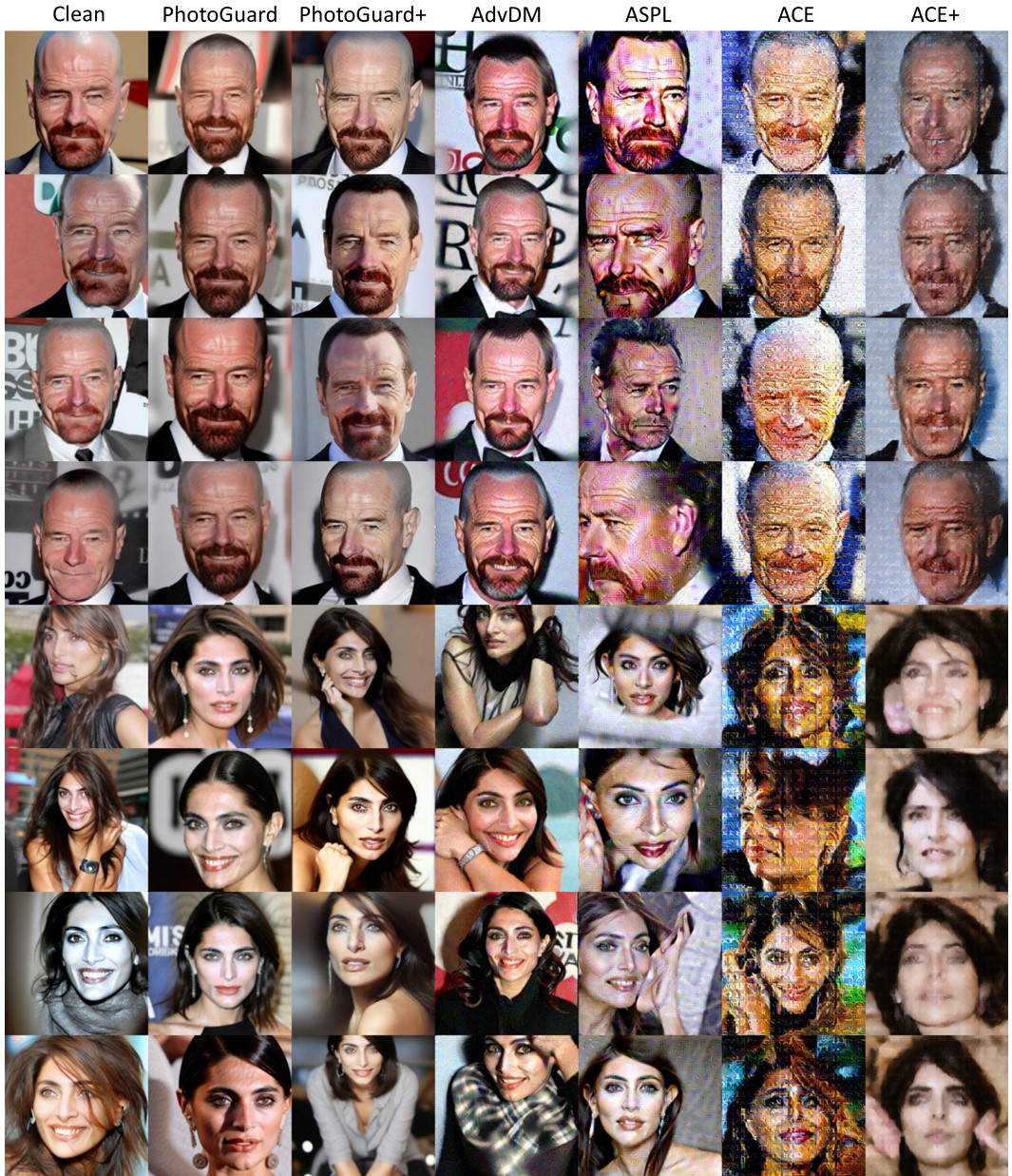

Figure 15: Output images of LoRA under different adversarial attacks. Two proposed methods outperform baseline methods.

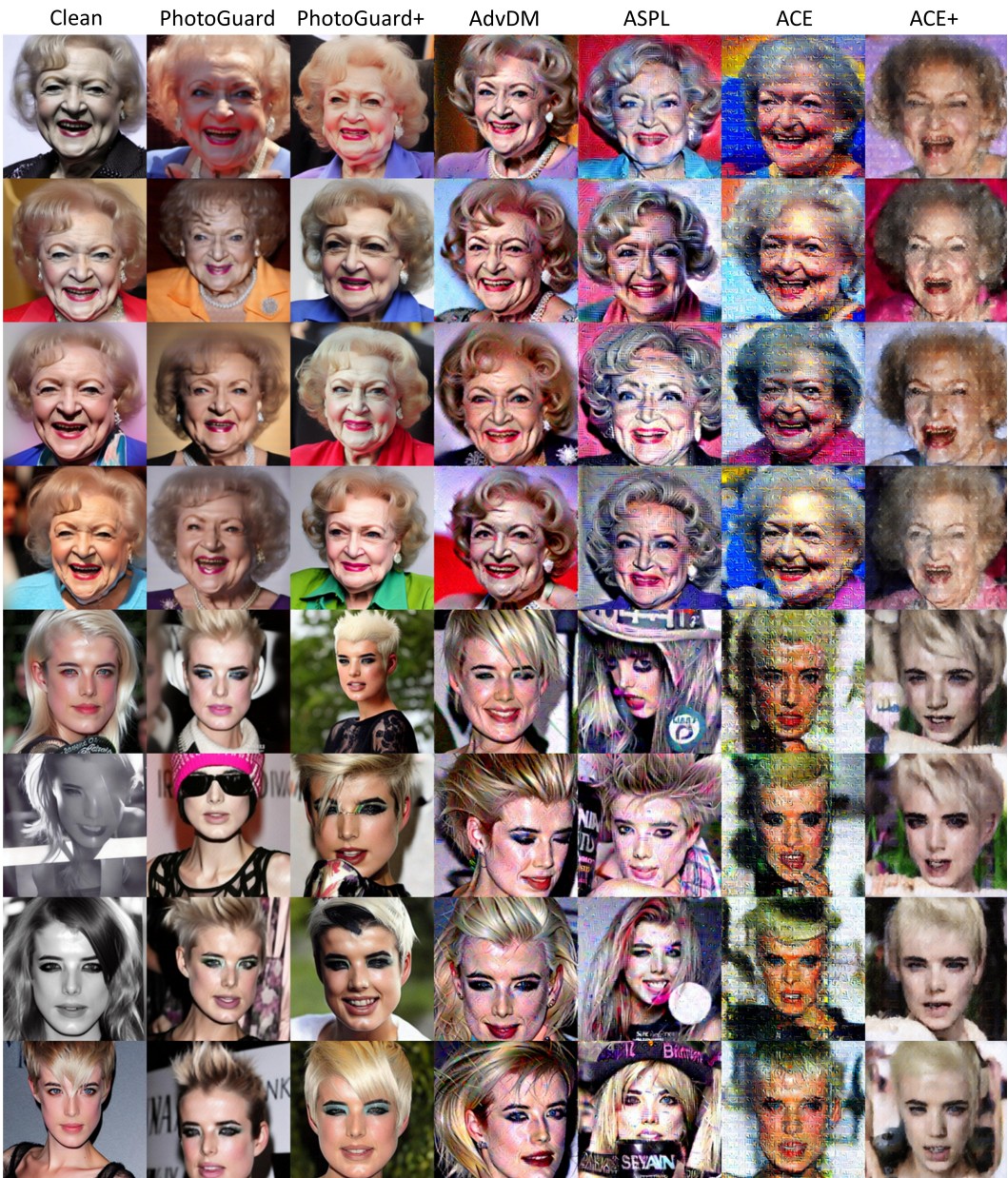

Figure 16: Output images of LoRA under different adversarial attacks. ACE and ACE+ outperform baseline methods. (Cond.)

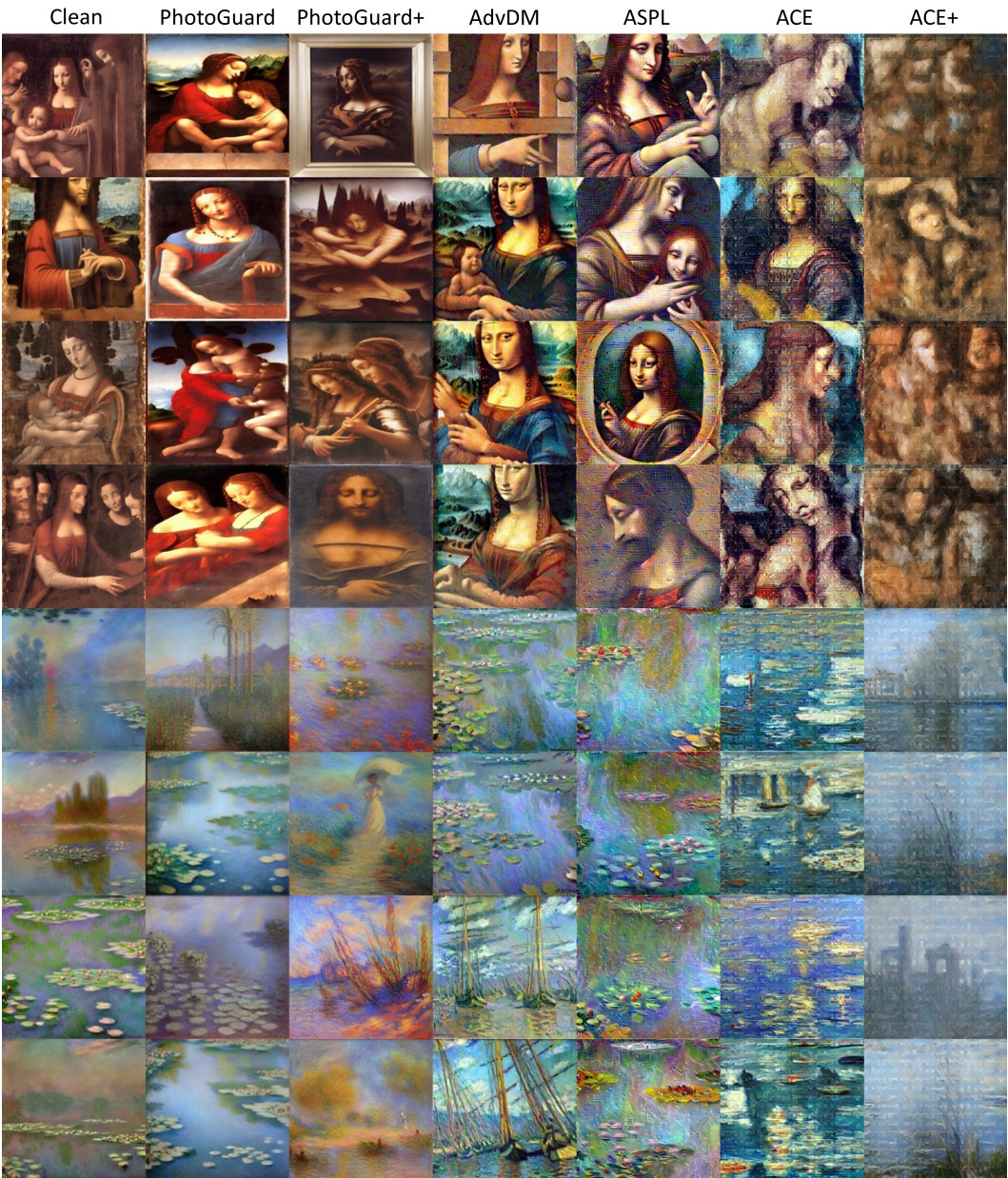

Figure 17: Output images of LoRA under different adversarial attacks. ACE and ACE+ outperform baseline methods. (Cond.)

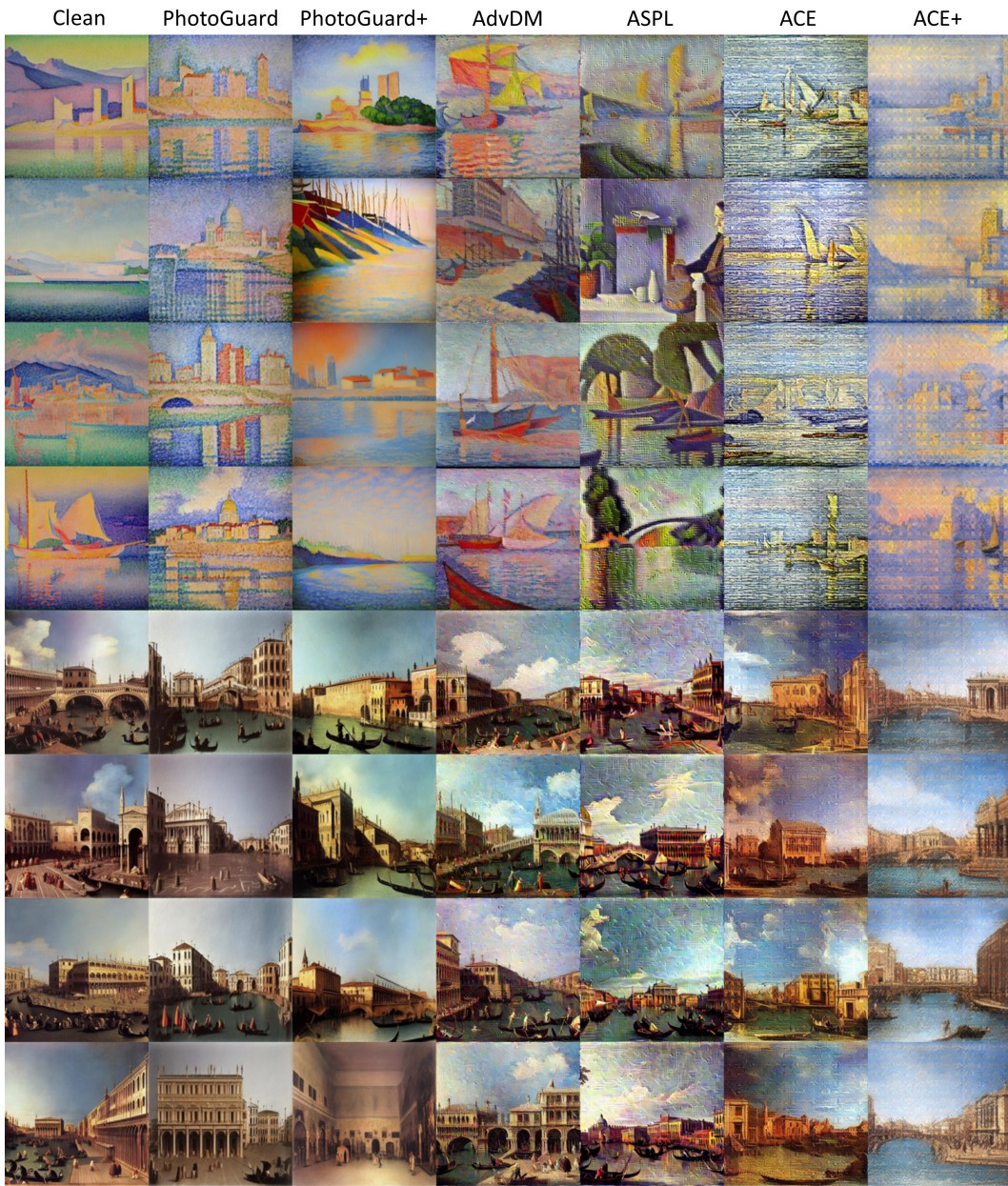

Figure 18: Output images of LoRA under different adversarial attacks. ACE and ACE+ outperform baseline methods. (Cond.)

