# OpenReview forum: "Targeted Attack Improves Protection against Unauthorized Diffusion Customization"
_ICLR.cc/2025/Conference — ICLR 2025 Spotlight_

### Official Review · Reviewer_v9HM · 2024-10-29

**Soundness:** 3
**Presentation:** 3
**Contribution:** 3
**Rating:** 8
**Confidence:** 4

**Summary:**

This paper focuses on protecting images from being exploited by sota T2I diffusion models via adversarial perturbation. Different from previous works that mainly utilize untargeted attacks, this work points out an interesting and novel perspective that targeted attacks works better. The experimental results prove the method's effectiveness. The author also provides an insight to explain the success of targted attack: the attack can lead to model to learn more consistent chaotic patterns, and mitigate the neutralization effect of untargeted perturbation from multiple samples.

**Strengths:**

- The idea is simple and interesting, and the proposed method is easy to implement.

- The discovery that targeted attacks on diffusion-based mimicry works better than untargeted ones is novel and interesting.

- This paper is written in a clear way, and the logic of this paper is easy to follow.

- The performance reported in the paper demonstrates the effectiveness of the proposed technique.

- This paper tries to provide some insights to explain the underlying reason of why targeted attacks works better than untargeted attacks. The reasons and explanations are intuitive and reasonable.

**Weaknesses:**

My concerns mainly lie in the experimental evaluation part of the paper. The authors seem to be not closely following the recent advances in adversarial attacks & defenses for protection against diffusion-driven mimicry/editting.

- Lack of recent attack baselines (in 2024) for comparison in the experiments, such as SDS, MetaCloak, Influence Watermarks, etc.

- The paper only focuses on latent diffusion models, while other models such as diffusion transformers are not evaluated. I suggest to conduct more experiments on state-of-the-art DiT models. Although they might not be tailored for personalization, some of them can be easily adopted for image editting tasks.

- The purification experiments does not include any purification methods specifically designed for diffusion mimicry, such as IMPRESS [1], GrIDPure [2], etc.

- I also have a concern on the superiority of targeted attacks claimed in this paper. In the analyses part (Section 5), the main idea is that as targeted attacks can lead to model to learn more consistent chaotic patterns, they mitigate the neutralization effect of untargeted perturbation from multiple samples. However, for image editting tasks and some state-of-the-art personalization methods, only one reference image is involved, and the above "neutralization effect" will not neccessarily happen. How can the hypotheses in Section 5 explain the empirical success of targeted attacks in these settings?

Minor Points:

- The claim in L144 that all existing protection are untargeted attacks is not accurate. For example, Glaze should be classified as a targeted attack. ASPL also proposes a targeted attack version.

- The description in L500 seems wrong. $\theta^{\prime}$ should be the customized diffusion model.

The reviewer will engage actively in the discussion and adjust the rating according to the rebuttal.

[1]: Cao et al. IMPRESS: Evaluating the Resilience of Imperceptible Perturbations Against Unauthorized Data Usage in Diffusion-Based Generative AI. NeurIPS 2023.

[2]: Zhao et al. Can Protective Perturbation Safeguard Personal Data from Being Exploited by Stable Diffusion? CVPR 2024.

**Questions:**

- In algorithm 1 Line 5-7, why are you optimizing the diffusion model parameters as well? Seems that these details are not included in Eq. (4)-(5).

- In table 1, what are the budgets of the baselines? Are they 4/255 or aligned with their own settings?

---

> ### Author Response · Authors · 2024-11-21
> **Response to Reviewer (1/2)**
>
> We thank the insightful review and would like to address the issue by points:
>
>
> [1] (W1: lack of recent baselines) We thank reviewers for mentioning new baselines and have included them in our related works in the updated version. We are still doing experiments on SDS and MetaCloak (Influence Watermark has no open-sourced implementation) and have not had their results ready due to the limited time. Also, we find that the mentioned three baselines may not be necessary baselines for our method, where we give the reasons one by one:
>
>
> (SDS) According to its abstract [A], SDS mainly focuses on doubling the speed of protection and reducing memory occupation by half without compromising its strength. This indicates that the focus of SDS is mainly on computational efficiency. While our focus is to improve the protection effectiveness and ACE shows clear superiority compared to AdvDM, which is the main baseline of SDS, we believe ACE could outperform SDS.
>
>
> (MetaCloak) MetaCloak mainly focuses on robustness to Gaussian transformation. According to Figure 2 in [B], its protection effectiveness is inferior to ASPL under no transformation scenarios. On the other hand, ACE outperforms ASPL clearly according to our experiments. Hence, we believe that ACE could outperform MetaCloak under no transformation scenarios. Also, it is possible that we can embed ACE in the objective term of MetaCloak to yield both good effectiveness and robustness.
>
>
> (Influence Watermark) Influence Watermark focuses on protection effectiveness. However, it has neither open-sourced implementation nor information for reproduction (even the adversarial budget is missing). Hence, we can only provide some insights which may help the reviewer to compare our method with it. Two metrics are shared on the same dataset for Influence Watermark and ACE: FDFR and ISM, which mean the failure ratio of generated portraits being detected as faces and the ratio of generated portraits being detected as the correct identity. Our FDFR is 1.0, that none generated portraits are detected as faces and ISM is NA, because none generated portraits are detected as faces with some identity. **This is the upper bound of protection effectiveness measured by these two metrics**. By contrast, the FDFR and ISM of Influence Watermark are 0.28-0.67 and 0.31-0.33, respectively [C]. While Influence Watermark does not provide the adversarial budget, ACE uses the lowest budget of all existing works, 4/255. Hence, we believe that ACE has sound advantages in effectiveness compared to Influence Watermark.
>
> Generally speaking, SDS and MetaCloak do not focus on improving the protection effectiveness, while Influence Watermark seems to be inferior to our method in protection effectiveness. Although these works contribute to the field from different perspectives, we believe ACE has its unique significance in effectiveness. We have updated this discussion on baseline selection in our Appendix A (L788-L797) to make it acknowledged.
>
> [2] (W2: experiments on DiT) We conducted new experiments on Stable Diffusion 3, which is the state-of-the-art DiT-based diffusion model. The details are given in Appendix B.4 (L1026-L043). We find that ACE can still degrade diffusion customization on SD3, but in a different way: Instead of degrading the image quality, ACE makes images irrelevant to the base training dataset. As shown in Figure 7 in our new draft, images generated by LoRA under ACE cannot generate portraits with the same person in the training dataset. ACE is the first attack that proves to work on Stable Diffusion 3.
>
>
> [3] (W3: specifical purification) First, our robustness experiments cover more purification methods than the union of our baselines e.g. DiffPure. We believe this is enough to show the robustness of our methods. Second, specifical purification is a novel challenge to the robustness of all protections. To overcome this requires also specifical efforts in designing the adversarial constraints and patterns, which could be a different topic to study. Notably, Good Protection=Effective Objective + Robust Pattern. While our focus is mainly on the objective, we would like to leave countering specifical purification to future work. Finally, we understand the reviewer’s concern in the potential overclaim of our real-world effectiveness. To clarify this, we add the following statement in our limitation (Appendix D, L1200-L1209)

---

> > ### Author Response · Authors · 2024-11-21
> > **Response to Reviewer (2/2)**
> >
> > *In addition, we follow current works to use l2-distance to constrain our adversarial perturbations, which makes comparisons fair. This design, however, **could be vulnerable** to some new advanced purification methods, for example, IMPRESS and GrIDPure. We believe this could be overcome by rectifying the constraint and designing patterns that hide our protection deeper in the original image. However, this is out of the scope of this paper which focuses on optimizing the attack objective. We believe our stronger objective can contribute to real-world protection in combination with a potential robust constraint and pattern, where we leave to the future work. Also, our robustness experiments show that our method has robustness against general purification, keeping its essential utility. Hence, we leave the improvement of the robustness against advanced purification to the future work.*
> >
> >
> >
> > [4] (W4: explanation of effectiveness on image editing) The success of our targeted attack works could be attributed to two points: (1) the pattern is chaotic and (2) patterns of different images are consistent. Our hypothesis is only about the (2) and mainly explains why ACE performs significantly better than baselines on degrading fine-tuning. For image editing the empirical success of ACE mostly relies on the carefully selected chaotic pattern in the target (point (1)). In other words, we do not extend the hypothesis to image editing. To clarify this, we rectify our expression at the beginning of our hypothesis in L479 of our new draft.
> >
> >
> > [5] (W5 & W6: claim on untargeted attacks & clerical mistake at L500) We feel sorry for the inaccurate expressions and have fixed them in our new draft.
> >
> >
> > [6] (Q1: why training the parameters) As mentioned in L235, we followed ASPL to find that this term could improve the performance against fine-tuning based customization while keeping effectiveness against image editing. This is the reason why we added this term.
> >
> >
> > [7] (Q2: adversarial budget) Yes, all baselines use the budget of 4/255 to make comparisons fair.
> >
> > Again, we thank the reviewer for the insightful review. If you have further questions, feel free to contact us.
> >
> >
> > Reference:
> >
> >
> > [A] Xue, Haotian, et al. "Toward effective protection against diffusion-based mimicry through score distillation." The Twelfth International Conference on Learning Representations. 2023.
> >
> >
> > [B] Liu, Yixin, et al. "MetaCloak: Preventing Unauthorized Subject-driven Text-to-image Diffusion-based Synthesis via Meta-learning." Proceedings of the IEEE/CVF Conference on Computer Vision and Pattern Recognition. 2024.
> >
> >
> > [C] Liu, Hanwen, Zhicheng Sun, and Yadong Mu. "Countering Personalized Text-to-Image Generation with Influence Watermarks." Proceedings of the IEEE/CVF Conference on Computer Vision and Pattern Recognition. 2024.

---

> > > ### Comment · Reviewer_v9HM · 2024-11-24
> > > **Thank you for your response.**
> > >
> > > Dear authors, thank you for your responses. I believe you've done a good job rebuttal as it comprehensively addressed my concerns. It's surprising to see the results on SD3 that the model trained on protected images generates completely different faces rather than chaotic patterns. It would be interesting to add more analyses and discussions on this aspect in your revision, which would be insightful for future works. I've raised my score to 8.

---

> > > > ### Author Response · Authors · 2024-11-25
> > > > **Response to Reviewer**
> > > >
> > > > We sincerely thank the reviewer for raising the score and recognizing the contribution of this work. We also agree that it is valuable to further investigate the behavior of ACE on MMDiTs such as SD3. We will explore this further and include additional discussions in our next draft.

---

### Official Review · Reviewer_p5eN · 2024-10-31

**Soundness:** 2
**Presentation:** 3
**Contribution:** 2
**Rating:** 8
**Confidence:** 3

**Summary:**

This paper discusses the use of targeted adversarial attacks to improve protection against unauthorized diffusion customization in image generation models. Traditional protections using untargeted attacks are not effective enough, so the authors propose a method called Attacking with Consistent score-function Errors (ACE). ACE significantly degrades the quality of customized images by introducing targeted errors, making unauthorized customization less viable. The paper validates ACE's effectiveness through extensive experiments and provides insights into the mechanisms of attack-based protections, setting a new benchmark in the field.

**Strengths:**

1. Relevant and timely topic
2. Clear presentation of ideas
3. Thorough evaluation

The authors address the important and timely topic of protecting diffusion models from unauthorized fine-tuning on specific images. To overcome limitations in existing approaches, they shift the focus from untargeted to targeted attacks, introducing a novel protection method called ACE. The authors provide a clear explanation of the design and rationale behind their method, which is commendable. Additionally, they conduct extensive evaluations that demonstrate the method's effectiveness. Overall, this paper is well-written and easy to follow.

**Weaknesses:**

1. **Limited Technical Contribution**: The proposed method primarily relies on existing adversarial attacks, such as PGD, to mislead the model toward a predefined target. However, all of the techniques applied are directly taken from prior works, with minimal novel adaptation or extension.

2. **Lack of Evaluation on SD3**: The evaluations are conducted exclusively on SD versions 1.4, 1.5, and 2.1, which are somewhat outdated. Although testing on SD3 could provide valuable insights, it’s worth noting that SD3 is not open-source and lacks support for customization pipelines like LoRA or DreamBooth, which are central to this paper’s focus on protecting against unauthorized customization. Including a discussion of this limitation and potential future directions for adapting the method to newer models could strengthen the paper.

3. **Potential Bias in Selected Evaluation Images**: The authors selected images from two datasets to evaluate the performance of various customization and protection methods. However, it's unclear if these images were part of the original SD models’ training set, which could introduce bias. It would be helpful if the authors could clarify whether they verified that the selected images were not in the SD models’ training data. If verification wasn’t conducted, acknowledging this limitation would be beneficial.

4. **Additional Baseline for Comparison**: While the authors include several baselines, an important reference—**"Adversarial Perturbations Cannot Reliably Protect Artists from Generative AI"**—is missing. Including this work would allow for a more comprehensive comparison. Specifically, a discussion on how the proposed method compares to or improves upon the reliability and effectiveness of the techniques discussed in this paper would provide valuable context.

**Questions:**

1. Did the authors verify that the selected evaluation images were not part of the original SD models' training datasets to minimize potential bias?

2. Please consider including the suggested work as an additional baseline for evaluating the proposed method.

---

> ### Author Response · Authors · 2024-11-21
> **Response to Reviewer**
>
> We thank the insightful review and would like to address the issue by points:
>
> (1) (lack of novelty): First we would like to point out that PGD attack has nearly become a standard approach for many modern adversarial attacks. Many works in this field adopt PGD as their base attack method [1][2][3][4]. In the field of adversarial attack against diffusion models, our method is fundamentally different from previous works. Though the idea of setting a target for attack is already proposed by [1], the performance of targeted attack still lags behind untargeted ones. We’re the first to advance the performance of targeted attack to SOTA level, even pushing the ASR (Attack Success Rate) on CelebA-HQ to the upper bound: 100\%.
> Our work also differs from other targeted-attack works from the formulation. Previous works [1][3] tried to pull the flow of diffusion models to a predefined poisoned flow. ACE, in contrast, tried to pull the flow towards a fixed, time-invariant target, which cannot be interpreted as a legit flow. Instead, we formulated this as a consistent score function error over time. This novel view gives targeted attack strong performance, outperforming untargeted ones and reaches the SOTA level.
>
>
> (2) (SD3) We thank the reviewer for bringing SD3 out. As the reviewer mentioned,  LoRA on SD3 is not fully supported due to its structure difference to SD1/2. But still as the reviewer requested, we conduct experiments to empirically show that ACE can adapt to SD3 under the LoRA setup. The details are given in Appendix B.4 (L1026-L1043). We find that ACE can still degrade diffusion customization on SD3, but in a different way: Instead of degrading the image quality, ACE makes images irrelevant to the base training dataset. As shown in Figure 7 in our new draft, images generated by LoRA under ACE cannot generate portraits with the same person in the training dataset. ACE is the first attack that proves to work on Stable Diffusion 3. This behavior difference is intriguing, and possibly due to the architecture difference between SD3 and SD1.x.
>
> (3) (potential bias) We agree that the Stable Diffusion may already integrate our evaluation data into the training set, as the datasets we use are accessible on the Internet. However, our dataset selection follows the setup of baselines. While some of them have been applied widely, we believe this potential bias does not constitute serious threats to our performance evaluation. Also, Membership Inference Attack (MIA) on large, commercial diffusion models is very hard to conduct, and it’s still an open and challenging problem for academia [5]. Conducting such validation is far beyond the focus of this paper. Finally, we have some positive samples drawn from the artists community, where the attack and evaluation is conducted by the artists on their private artworks. However the artist does not give us the consent to release them. We’ll acknowledge the miss of validation in our next revision.
>
>
> (4) (noisy-upscaling) We have discussed with the authors of ‘Adversarial Perturbations Cannot Reliably Protect Artists from Generative AI’ as soon as it is public. They evaluate ACE on Noisy-Upscaling. Compared to other watermarks like Mist-v1 and Glaze, ACE still survives Noisy-Upscaling to some extent that the produced images include some chaotic contents, for example, mixed candle bodies and flame. Considering that most of the watermarks have been broken by noisy-upscaling, ACE is the only one that keeps some effectiveness. However, due to the double-blind rule, we cannot share the image example here because it could be used to detect our paper.
>
>
> In addition, while our focus is mainly on an effective objective, we just align our robustness experiments to our baselines. We agree that there are specifical purification methods that raise true threats to protection methods like ACE. It is the common challenge of the field, while we believe methods to counter them could be an independent topic to study. Therefore, we would like to leave it to future work.
>
>
> Questions:
>
>
> [1] Please see Weakness (3).
>
> [2] Please see Weakness (4).
>
>
> Again, we thank the reviewer for the insightful review. If you have further questions, feel free to contact us.
>
>
>
>
> [1] : Van Le, Thanh, et al. "Anti-dreambooth: Protecting users from personalized text-to-image synthesis."
>
> [2]: Ahn, Namhyuk, et al. "Imperceptible Protection against Style Imitation from Diffusion Models."
>
> [3]: Liang, Chumeng, et al. "Adversarial example does good: Preventing painting imitation from diffusion models via adversarial examples."
>
> [4]: Chen, Yipu, Haotian Xue, and Yongxin Chen. "Diffusion Policy Attacker: Crafting Adversarial Attacks for Diffusion-based Policies."
>
> [5]: Dubiński, Jan, et al. "Towards more realistic membership inference attacks on large diffusion models."

---

> ### Author Response · Authors · 2024-11-28
> **Kindly request for further discussion**
>
> As the rebuttal period draws to a close, we kindly request your engagement in the discussion.
> We give further discussion about our technical contribution to this field. We also conduct experiments on SD3 to show our effectiveness. Notably, **we’re the first attack to show effectiveness on SD3**. We also discuss the effect of noise-upscaling and the potential bias of our training set.
> Please don’t hesitate to contact us if you have any further questions. We would sincerely appreciate it if you could consider engaging in further discussions.

---

> > ### Comment · Reviewer_p5eN · 2024-11-28
> >
> > Thanks to the authors have addressed most of my concerns, and I have raised the score.

---

> > > ### Author Response · Authors · 2024-11-28
> > > **Thank you for raising the score**
> > >
> > > We are sincerely grateful for your consideration. Thank you for raising the score!

---

### Official Review · Reviewer_3gvL · 2024-11-01

**Soundness:** 3
**Presentation:** 3
**Contribution:** 3
**Rating:** 8
**Confidence:** 4

**Summary:**

This paper proposes using targeted adversarial attacks to prevent unauthorized customized fine-tuning of LDM. The motivation is straightforward with a clear method design. The proposed ACE and ACE+ demonstrate a significant reduction in generation quality compared to baseline methods while maintaining an acceptable computational cost. The authors also conducted experiments to assess the robustness and transferability of ACE.

**Strengths:**

1. This is a well-written paper with solid experiments and analysis. The proposed ACE and ACE+ demonstrate superior performance against baseline methods.
2. The proposed method is straightforward and effective, with clear explanations for each step.

**Weaknesses:**

Given the detailed and solid experiments presented in this paper, I have no queries regarding the need for more ablation experiments. However, I do have concerns regarding the technical contributions and practical settings.

1. While this paper employs targeted adversarial attacks to prevent unauthorized customized fine-tuning, it is noted that this approach resembles Glaze [1], which also utilizes targeted style transfer to safeguard against style mimicry. Can you discuss the core difference between your technical contributions?

2. It is intriguing to investigate whether we should adopt ACE to all the protected images to achieve such protection results. What would be the impact on protection performance if the attacked ratio (the proportion of protected images) is reduced?

**Questions:**

My questions have been listed above. I'm willing to increase my score if my concerns are addressed.

---

> ### Author Response · Authors · 2024-11-21
> **Response to Reviewer**
>
> We thank the insightful review and would like to address the issue by points:
>
>
> [1] (Difference from Glaze) Unfortunately, Glaze does not have an official open sourced implementation and the demonstration in the paper does not give the details on the components of Stable Diffusion being used and how to use them to apply Glaze. However, based on the information we are acknowledged, we can still conclude two clear technical differences between ACE and Glaze:
>
>
> a) The objectives are different. ACE minimizes the distance between diffusion predicted latent noise and a fixed latent. We have both VAE and UNet involved. By contrast, Glaze minimizes the feature distance between the protected image and the target image in a feature representation space. It seems that the space is the VAE latent space of Stable Diffusion in their implementation. Only VAE is used in this process.
>
>
> b) The target selection is different. ACE exploits the latent of a chaotic-pattern image as the target. Glaze uses the style-transfered version of the original image as the target image.
>
>
> Generally speaking, two of our core contributions, a new objective and the target selection, are both novel compared to Glaze. Although ACE and Glaze are both called targeted attacks, we believe their technical cores are very different. We have added an extra section in Appendix D (L1244-L1264) for more details.
>
>
> In addition, we would like to note that Glaze does not use normal l2-constraint, thus making it impossible to do fair comparisons with all other baselines and our method. Hence, we follow previous works to not include Glaze in our baselines.
>
>
> [2] (performance of partial attacks) This is an interesting topic to investigate, while we believe this is a common issue of all protection methods rather than the specifical issue of ACE. Hence, we believe this is out of the scope of our core contributions on improving the effectiveness of protection methods. Despite this, we conducted a simple but intuitive experiment on SD3 to provide some insights. We merge the clean portrait dataset of person A and a protected portrait dataset of person B. We fine-tuned SD3 LoRA on this merged dataset and found that the fine-tuned LoRA can only generate the portraits of person A. This indicates that LoRA may ignore the protected images and only learn the unprotected images in this scenario. However, due to the time limit, we would like to leave the further exploration to future work.
>
>
> Again, we thank the reviewer for the insightful review. If you have further questions, feel free to contact us.

---

> > ### Comment · Reviewer_3gvL · 2024-11-22
> >
> > Thanks for the response. My first question regarding the comparison to Glaze has been resolved. For the second question, what if portrait B is partially protected by your ACE? Discussing this problem will be interesting and help comprehend the robustness of your ACE, which I consider crucial for not only your manuscript but also for current protection techniques. Nevertheless, I believe this paper provides a solid and comprehensive analysis, and I've raised my score to 8.

---

> > > ### Author Response · Authors · 2024-11-22
> > > **Response to Reviewer**
> > >
> > > We thank the reviewer for raising the score and recognize the contribution of this work. We also agree that it’s valuable to investigate the situation when portrait B is partially protected. We’ll try to investigate and post this if the rebuttal time limit allows.

---

### Official Review · Reviewer_YYfv · 2024-11-04

**Soundness:** 3
**Presentation:** 3
**Contribution:** 2
**Rating:** 6
**Confidence:** 3

**Summary:**

This paper proposes a targeted attack method for the protection against unauthorized diffusion customization. Extensive experiments validate the effectiveness of the targeted attack compared to other baselines. The paper also proposes an explanation for the effectiveness of targeted attacks.

**Strengths:**

1. The paper first proposes the targeted attacks for the protection against diffusion customization, and validates the effectiveness of targeted attacks.

2. The paper proposes an explanation for the superiority of targeted attacks, which may help understand attack-based protection.

3. The experiments on transferability and robustness also validate the effectiveness.

**Weaknesses:**

There are two main concerns.

1. Did the authors try some specifically purification methods for the protection? e.g., the method in [1]. Can the method purify the perturbations?

2. The authors may need to compare the added loss term alone in the ACE+ loss with the proposed ACE loss. The added loss in ACE+ is also targeted attack. Experimental comparisons are needed between it and the ACE loss.



[1] Bochuan Cao et al. IMPRESS: Evaluating the Resilience of Imperceptible Perturbations Against Unauthorized Data Usage in Diffusion-Based Generative AI, 2023.

**Questions:**

Please see the Weaknesses part.

---

> ### Author Response · Authors · 2024-11-21
> **Response to Reviewer**
>
> We thank the insightful review and would like to address the issue by points:
>
>
> [1] (specifical purification) First, our robustness experiments cover more purification methods than the union of our baselines e.g. DiffPure. We believe this is enough to show the robustness of our methods. Second, specifical purification is a novel challenge to the robustness of all protections. To overcome this requires also specifical efforts in designing the adversarial constraints and patterns, which could be a different topic to study. Notably, Good Protection=Effective Objective + Robust Pattern. While our focus is mainly on the objective, we would like to leave countering specifical purification to future work. Finally, we understand the reviewer’s concern in the potential overclaim of our real-world effectiveness. To clarify this, we add the following statement in our limitation (Appendix D, L1200-L1209)
>
>
> In addition, we follow current works to use l2-distance to constrain our adversarial perturbations, which makes comparisons fair. This design, however, **could be vulnerable** to some new advanced purification methods, for example, IMPRESS and GrIDPure. We believe this could be overcome by rectifying the constraint and designing patterns that hide our protection deeper in the original image. However, this is out of the scope of this paper which focuses on optimizing the attack objective. We believe our stronger objective can contribute to real-world protection in combination with a potential robust constraint and pattern, where we leave to the future work. Also, our robustness experiments show that our method has robustness against general purification, keeping its essential utility. Hence, we leave the improvement of the robustness against advanced purification to the future work.
>
>
> [2] (added loss in ACE+) The sole added loss in ACE is the PG in our baselines. We have listed its performance in Table 1, where it does not seem to be competitive compared to other baselines and our methods. Additionally, we add ACE+ mainly to show a variant with comparable performance to ACE and different patterns in the output images, which provides more choices for real-world practice. The added loss mainly serves to change the pattern of ACE rather than helping ACE+’s outperform ACE. Hence, we believe no extra ablation studies are needed.
>
>
> Again, we thank the reviewer for the insightful review. If you have further questions, feel free to contact us.

---

> ### Author Response · Authors · 2024-11-28
> **Kindly request for further discussion**
>
> Dear Reviewer,
>
> We highly appreciate your constructive review and comprehend your concerns on our robustness to specific purification methods as well as ablation studies. To address these concerns, we updated our paper draft and provided a detailed explanation above. As the rebuttal period draws to a close, we sincerely look forward to your further ideas on these concerns and kindly request your engagement in the discussion. Notably, our newly-conducted experiment shows that **our method is the first attack to show effectiveness on SD3** (See Appendix B.4, L1026-L1043). We also want to hear your advice on the updated content. We would sincerely appreciate it if you could consider engaging in further discussions.

---

> > ### Author Response · Authors · 2024-11-30
> > **Kindly request for further discussion**
> >
> > Dear Reviewer,
> >
> > As the rebuttal period draws to a close, we sincerely look forward to have your further discussion. We address your concern about the purification methods. Reviewr v9HM has also acknowledged our effort in examining purification methods and raise the score. We also address your concern about the added ACE loss, which is the PG in our baselines. We also want to hear your advice on the updated content. We would sincerely appreciate it if you could consider engaging in further discussions and adjusting the score.

---

> > > ### Author Response · Authors · 2024-12-02
> > > **Kindly request for further discussion**
> > >
> > > Dear Reviewer,
> > >
> > > As the rebuttal period is very near to the end, we sincerely look forward to have your further discussion. We would really appreciate it if you could provide more feedback and advice.

---

### Author Response · Authors · 2024-11-21
**Updates in Response to Reviews**

We thank all reviewers for the insightful reviews and have accordingly updated a new draft of our paper. We list the updated points, noted as blue text in our new draft, as follows:

[1] LoRA experiments on Stable Diffusion 3, one of the state-of-the-art DiT-based diffusion models. (Appendix B.4).

[2] References to some new attacks and purification methods and discussion on baseline selection (Appendix A).

[3] A statement about advanced purification methods and the real-world effectiveness of our methods (Appendix C).

[4] Clarification of the core differences between our method and other targeted attacks (Appendix B.5).

[5] Clarification of the domain of our hypothesis (Section 5).

[6] Fix some inaccurate expressions in Section 2 and Section 4.

---

### Meta-Review · Area_Chair_WWq8 · 2024-12-17

**Metareview:**

The paper proposes a method to prevent unauthorized diffusion customization. Rather than an untargeted attack, it proposes a targeted attack in the protected image that will show a designated texture or patterns after diffusion modification. It also includes valuable discussion on the proposed method's effectiveness. The paper is well-written and easy to follow. The experimental results show the effectiveness of the targeted attack. Please remember to incorporate some of the reviewer's concerns in the main text as well. I recommend accepting this work.

**Additional Comments On Reviewer Discussion:**

The author did great work addressing all reviewer's concerns and adding additional experiments/details/clarification in the appendix. All reviewers are happy about the rebuttal.

---

### Decision · Program_Chairs · 2025-01-22

Accept (Spotlight)